# Melatonin Modulates the Antioxidant Defenses and the Expression of Proinflammatory Mediators in Pancreatic Stellate Cells Subjected to Hypoxia

**DOI:** 10.3390/antiox10040577

**Published:** 2021-04-08

**Authors:** Matias Estaras, Manuel R. Gonzalez-Portillo, Remigio Martinez, Alfredo Garcia, Mario Estevez, Miguel Fernandez-Bermejo, Jose M. Mateos, Daniel Vara, Gerardo Blanco-Fernández, Diego Lopez-Guerra, Vicente Roncero, Gines M. Salido, Antonio Gonzalez

**Affiliations:** 1Institute of Molecular Pathology Biomarkers, University of Extremadura, 10003 Caceres, Spain; meh@unex.es (M.E.); ramonglezpor@unex.es (M.R.G.-P.); gsalido@unex.es (G.M.S.); 2Department of Animal Health, Veterinary Faculty, University of Extremadura, 10003 Caceres, Spain; remimar@unex.es; 3Department of Animal Production, CICYTEX-La Orden, 06187 Badajoz, Spain; alfredo.garcia@juntaex.es; 4IPROCAR Research Institute, Food Technology, University of Extremadura, 10003 Cáceres, Spain; mariovet@unex.es; 5Department of Gastroenterology, San Pedro de Alcantara Hospital, 10003 Caceres, Spain; mfbermejo@unex.es (M.F.-B.); josemaria.mateos@salud-juntaex.es (J.M.M.); daniel.vara@salud-juntaex.es (D.V.); 6Hepatobiliary-Pancreatic Surgery and Liver Transplant Unit, Infanta Cristina Hospital, 06080 Badajoz, Spain; gerardoblanco@unex.es (G.B.-F.); diego.lopezg@salud-juntaex.es (D.L.-G.); 7Unit of Histology and Pathological Anatomy, Veterinary Faculty, University of Extremadura, 10003 Caceres, Spain; roncero@unex.es

**Keywords:** cell viability, glutathione, hypoxia, inflammation, melatonin, pancreatic stellate cells, reactive oxygen species

## Abstract

Pancreatic stellate cells (PSC) play a major role in the formation of fibrotic tissue in pancreatic tumors. On its side, melatonin is a putative therapeutic agent for pancreatic cancer and inflammation. In this work, the actions of melatonin on PSC subjected to hypoxia were evaluated. Reactive oxygen species (ROS) generation reduced (GSH) and oxidized (GSSG) levels of glutathione, and protein and lipid oxidation were analyzed. The phosphorylation of nuclear factor erythroid 2-related factor (Nrf2), nuclear factor kappa-light-chain-enhancer of activated B cells (NF-kB), and the regulatory protein nuclear factor of kappa light polypeptide gene enhancer in B-cells inhibitor-alpha (IκBα) was studied. The expression of Nrf2-regulated antioxidant enzymes, superoxide dismutase (SOD) enzymes, cyclooxygenase 2 (COX-2), interleukin-6 (IL-6) and tumor necrosis factor-α (TNF-α) were also studied. Total antioxidant capacity (TAC) was assayed. Finally, cell viability was studied. Under hypoxia and in the presence of melatonin generation of ROS was observed. No increases in the oxidation of proteins or lipids were detected. The phosphorylation of Nrf2 and the expression of the antioxidant enzymes catalytic subunit of glutamate-cysteine ligase, catalase, NAD(P)H-quinone oxidoreductase 1, heme oxygenase-1, SOD1, and of SOD2 were augmented. The TAC was increased. Protein kinase C was involved in the effects of melatonin. Melatonin decreased the GSH/GSSG ratio at the highest concentration tested. Cell viability dropped in the presence of melatonin. Finally, melatonin diminished the phosphorylation of NF-kB and the expression of COX-2, IL-6, and TNF-α. Our results indicate that melatonin, at pharmacological concentrations, modulates the red-ox state, viability, and the expression of proinflammatory mediators in PSC subjected to hypoxia.

## 1. Introduction

Activated pancreatic stellate cells (PSC) play a major role in the fibrotic process that develops in the pancreas in inflammation and cancer [1]. Fibrosis acts as a barrier against antitumor agents and, thereby, represents a factor of resistance in cancer [2]. In the three-dimensional matrix that is built up by the fibrotic components, cells establish specific interactions and cooperate between each other. This leads to tumor growth in parallel with resistance against chemo- and radiotherapy [3,4].

Hypoxia is a condition that is present in the vast majority of tumors and is a consequence of the fast proliferation and accumulation of cells within the growing tissue [5]. The cells comprising the mass exhibit adaptation to the low oxygen availability and set up different mechanisms that will help them survive. These changes allow the growth of the tumor [6].

In a recent work, we have shown that PSCs exhibited adaptation to hypoxia and were able to proliferate [7]. This is a major change that takes place in PSCs, which might contribute to the perpetuation of the fibrosis cycle inside a malignant tissue and could also contribute to inflammation [8]. Therefore, modulation of the growth of fibrotic tissue within tumors might be of outstanding importance in the treatment of inflammation and cancer.

Melatonin is the product of the pineal gland and is subjected to a rhythmic production and secretion during the dark phase of the day [9]. Melatonin depicts antioxidant properties on the exocrine pancreas [10]. Mainly, its mechanisms of action involve detoxification of free radicals by direct electron donation and by modulation of antioxidant defenses [11]. To cite some examples, melatonin reverted glutathione peroxidase activity in cerulein-induced acute pancreatitis [12], increased the levels of superoxide dismutase (SOD) activity and of reduced glutathione (GSH) in plasma and macrophages of animals infected with *Trypanosoma cruzi* [13], increased glutathione reductase content in liver, kidney, heart, and testis tissues [14], and induced the activation of the nuclear factor erythroid 2-related factor (Nrf2) and the antioxidant-responsive element, leading to related antioxidant enzymes, in pancreatic acinar cells [15]. In addition, melatonin exhibits a wide range of anticancer activities as for example in lung cancer [16], liver cancer [17], colorectal cancer [18], or pancreatic cancer [19]. Effects of melatonin against inflammation in the pancreas have also been shown [20]. With regard to PSC, we have shown previously that melatonin decreases the viability of this cell type under normoxia [21,22].

The major actions of melatonin are carried out through the activation of membrane-bound receptors [23,24]. However, direct actions of melatonin have also been proposed [22]. Interestingly, PSCs do not exhibit such membrane-located receptors for melatonin [21,22]. Therefore, the means by which melatonin exerts its actions on PSC physiology is intriguing. As mentioned above, PSCs are able to proliferate under hypoxia. To some extent, PSCs exhibited antioxidant responses that might underlie the mechanisms by which cells adapt to the low availability of oxygen (O_2_) [7]. Hence, bearing in mind the antiproliferative actions of melatonin and the contribution of PSC to inflammation and cancer growth, the study of the mechanisms of action of melatonin on PSC physiology needs further consideration.

In the present study, we have continued our previous work and have investigated the effects of melatonin on PSC subjected to hypoxia. We were interested in clarifying the mechanisms by which the indoleamine could modulate PSC physiology in order to demonstrate its therapeutic potential in the treatment of pancreatic inflammation and cancer.

## 2. Materials and Methods

### 2.1. Chemicals

Collagenase was purchased from Worthington Biochemical Corporation (Labclinics, Madrid, Spain). Cell Lytic for cell lysis and protein solubilization, crystal violet, hydrogen peroxide (H_2_O_2_), N-ethylmaleimide, oxidized glutathione, O-phthalaldehyde, protease inhibitor cocktail (Complete, EDTA-free), reduced glutathione, thapsigargin, and Tween^®^-20 were obtained from Sigma Chemicals Co. (Madrid, Spain). CM-H_2_DCFDA (5-(and-6)-chloromethyl-2′,7′-dichlorodihydrofluorescein diacetate acetyl ester), fetal bovine serum (FBS), Hank’s balanced salts (HBSS), horse serum, and medium 199 were obtained from Invitrogen (Fisher Scientific Inc., Madrid, Spain). Polystyrene plates for cell culture and primers for RT-qPCR were purchased from Thermo Fisher Sci. (Madrid, Spain). Penicillin/streptomycin was purchased from BioWhittaker (Lonza, Basel, Switzerland). Bradford’s reagent, Tris/glycine/SDS buffer (10×), and Tris/glycine buffer (10×) were from Bio-Rad (Madrid, Spain). SignalFire™ ECL Reagent was obtained from Cell Signaling Technology (C-Viral, Madrid, Spain). Total antioxidant capacity (TAC) assay kit was obtained from BioVision (Deltaclon S.L., Madrid, Spain). Ro-31-8220 was purchased from Calbiochem (Sigma Chemicals Co., Madrid, Spain). The primary antibodies used in the study are listed in Table 1. The corresponding HRP-conjugated species-specific secondary antibody was employed. All other analytical grade chemicals used were obtained from Sigma Chemicals Co. (Madrid, Spain).

The primary antibodies listed were specific for each protein. The detection of the desired protein was carried out by Western blotting analysis, as described in the Methods section. Thermo Fisher (Madrid, Spain); Santa Cruz Biotechnology (Quimigen S.L., Madrid, Spain); Cell Signaling (C-Viral, Madrid, Spain).

### 2.2. Culture of Pancreatic Stellate Cells

To prepare the cultures of PSC, we employed methods described previously [22]. The pancreas was obtained from Wistar rat pups (3–5 days after birth). Animals were supplied by the animal house of the University of Extremadura (Caceres, Spain). A suspension of pancreatic cells was prepared by enzymatic digestion of the pancreas during 50 min at 37 °C with a Na-Hepes buffer (130 mM NaCl, 4.7 mM KCl, 1.3 mM CaCl_2_, 1 mM MgCl_2_, 1.2 mM KH_2_PO_4_, 10 mM glucose, 10 mM HEPES, 0.01% trypsin inhibitor (soybean), and 0.2% bovine serum albumin -pH 7.4 adjusted with NaOH-) supplemented with collagenase (30 Units/mL). Next, the tissue was subjected to gentle pipetting through tips of decreasing diameter to obtain a cell suspension. Thereafter, the suspension was centrifuged at 30× *g* for 5 min. at 4 °C, the supernatant was discarded, and the pellet was resuspended in culture medium, which was comprised of: Medium 199 supplemented with 4% horse serum, 10% FBS, a mixture of antibiotics (0.1 mg/mL streptomycin, 100 IU penicillin), and 1 mM NaHCO_3_. Small aliquots of cell suspension were then seeded on polystyrene plates for cell culture. The cells were grown in a humidified incubator with controlled temperature (37 °C) and CO_2_ (5%). With this procedure, an enriched culture of activated PSC with no contamination of other cell types is obtained [21,22]. Purity of the cultures was checked by determination of the expression of α-smooth muscle actin and of collagen type 1 (Appendix A), which are specific markers for activated PSC [25,26]. Confluence (90–95%) was reached after eight-ten days of culture. Different cell preparations were used in the studies.

### 2.3. Induction of Hypoxia

For induction of hypoxia, cells were incubated in a low O_2_ atmosphere employing an incubator chamber (Okolab; Izasa Scientific, Madrid, Spain) and following previous protocols [7]. Temperature (37 °C), humidity (90%), and air atmosphere (content of 1% O_2_/5% CO_2_/94% N_2_) were electronically controlled. The pH of the medium (7.30–7.35) did not change upon addition of drugs nor during the incubation periods.

### 2.4. Experimental Conditions for Melatonin Treatment

In the experimental procedures employed, separate batches of cells were incubated under hypoxia and in the presence of varying concentrations of melatonin (1000 µM, 100 µM, 10 µM, or 1 µM), or H_2_O_2_ (100 µM) or thapsigargin (Tps; 1 µM). For comparisons, separate batches of cells were incubated under hypoxia and in the absence of drugs (non-treated cells). Unless specifically stated, the incubation period lasted 4 h either in the absence or in the presence of drugs. In the experiments in which the PKC inhibitor Ro-31-8220 (5 µM) was used, cells were preincubated during 5 min in the presence of the inhibitor, prior to addition of the corresponding concentration of melatonin.

### 2.5. Determination of Reactive Oxygen Species Generation

Production of cytosolic reactive oxygen species (ROS) was detected following the manufacturer’s directions and using methods described previously [27]. Briefly, cells were detached and loaded with the fluorescent probe CM-H_2_DCFDA (10 µM). Cells were subjected to hypoxia and incubated with the indicated compounds for 1 h. Redox state of cells was monitored by measuring cellular fluorescence at 530 nm/590 nm (excitation/emission). Detection of fluorescence was performed using a plate reader (CLARIOstar Plus, BMG Labtech., C-Viral, Madrid, Spain). The experiments were carried out employing separate batches of cells, obtained from different preparations. Results are expressed as the mean increase of fluorescence in percentage ± SEM (*n*) with respect to cells that were subjected to hypoxia, but treated with no drug (non-treated cells) (*n* is the number of independent experiments).

### 2.6. Determination of Protein Carbonyls (Allysine)

After treatment, cells were washed with standard phosphate saline solution and were lysed for analysis. Protein carbonyls were determined following the methods described by Villaverde et al. [28], with slight modifications [29]. Five hundred µL of each sample were dispensed in 2 mL microtubes and treated with cold 10% trichloroacetic acid (TCA) solution. Each microtube was vortexed and then subjected to centrifugation at 600× *g* for 5 min at 4 °C. The supernatants were removed, and the pellets were incubated with the following freshly prepared solutions: 0.5 mL 250 mM 2-(N-morpholino) ethanesulfonic acid (MES) buffer pH 6.0 containing 1 mM diethylenetriaminepentaacetic acid (DTPA), 0.5 mL 50 mM ABA in 250 mM MES buffer pH 6.0, and 0.25 mL 100 mM NaBH3CN in 250 mM MES buffer pH 6.0. The tubes were vortexed and then incubated in water bath at 37 °C for 90 min. The samples were stirred every 15 min. After derivatization, samples were treated with a cold 50% TCA solution and centrifuged at 1200× *g* for 10 min. The pellets were then washed twice with 10% TCA and diethyl ether-ethanol (1:1). Finally, the pellet was treated with 6 M HCl and kept in an oven at 110 °C for 18 h until completion of hydrolysis. The hydrolysates were dried in vacuo in a centrifugal evaporator. The generated residue was reconstituted with 200 µL of milliQ water and then filtered through hydrophilic polypropylene GH Polypro (GHP) syringe filters (0.45 μm pore size, Pall Corporation, USA) for HPLC analysis.

The content of carbonyls was determined employing a Shimadzu “Prominence” HPLC apparatus (Shimadzu Corporation, Japan), equipped with a quaternary solvent delivery system (LC-20AD), a DGU-20AS on-line degasser, a SIL-20A auto-sampler, a RF-10A XL fluorescence detector, and a CBM-20A system controller. An aliquot (1 μL) from the reconstituted protein hydrolysates was injected and analyzed in the above mentioned HPLC equipment. AAS- ABA was eluted in a Cosmosil 5C18-AR-II RP-HPLC column (5 µm, 150 × 4.6 mm) equipped with a guard column (10 × 4.6 mm) packed with the same material. The flow rate was kept at 1 mL/min and the temperature of the column was maintained constant at 30 °C. The eluate was monitored with excitation and emission wavelengths set at 283 and 350 nm, respectively. Standards (0.1 μL) were run and analyzed under the same conditions. Identification of both derivatized semialdehydes in the FLD chromatograms was carried out by comparing their retention times with those from the standard compounds. The peak corresponding to allysine-ABA was manually integrated from FLD chromatograms and the resulting areas plotted against an ABA standard curve with known concentrations that ranged from 0.1 to 0.5 mM. The nmol of allysine per mg of protein were calculated. Results are expressed as percentage of change of allysine content ± SEM (*n*) with respect to non-treated cells under hypoxia (n is the number of independent experiments).

### 2.7. Analysis of Thiobarbituric-Reactive Substances

After treatment, cells were lysed for analysis. Malondialdehyde (MDA) and other thiobarbituric-reactive substances (TBARS) were measured in 200 µL samples of each treatment, by adding 500 µL thiobarbituric acid (0.02 M) and 500 µL trichloroacetic acid (10%), and following incubation during 20 min at 90 °C. After cooling, a 5 min centrifugation at 600× *g* was made, and the absorbance of supernatant was measured at 532 nm using a microplate reader. The mg of TBARS per L of sample were calculated. Results are expressed as percentage of TBARS ± SEM (n) with respect to non-treated cells under to hypoxia (n is the number of independent experiments).

### 2.8. Determination of Glutathione Levels

The changes in the levels of reduced (GSH) and oxidized (GSSG) glutathione were determined using methods described previously [21]. Cells were incubated with the different stimuli assayed. A spectrofluorimeter (Tecan Infinite M200, Grödig, Austria) was employed to measure the fluorescence of each sample at 350 nm/420 nm (excitation/emission), respectively. For quantification, standard curves of GSH and GSSG were used. Normalization was carried out based on the total protein concentration in each sample (Bradford, 1976). A standard curve was prepared using bovine serum albumin. The experiments were carried out employing batches of cells obtained from different preparations. Data show the mean increase in GSH/GSSG ratio expressed in percentage ± SEM (n) with respect to non-treated cells under hypoxia (n is the number of independent experiments).

### 2.9. Determination of Total Antioxidant Capacity

Total antioxidant capacity (TAC) was determined using a commercially available kit, following manufacturer’s directions, as described previously [29]. In brief, a working solution containing Cu^2+^, provided with the assay kit, was added to all standard and sample wells. Samples were then incubated during 1.5 h in the dark. Trolox standard curve was used for calibration, as described in the assay kit’s datasheet. A plate reader (CLARIOstar Plus, BMG Labtech., C-Viral, Madrid, Spain) was used to measure the absorbance (570 nm) of each sample. Results show the mean change of absorbance expressed in percentage ± SEM (n) with respect to non-treated cells incubated under hypoxia (n is the number of independent experiments).

### 2.10. Western Blotting Analysis

Western blotting analysis was employed for the determination of protein expression and/or phosphorylation, as described previously [30]. Bradford’s method was used for quantification of the protein content of lysates [31]. Protein lysates (15 µg/lane) of each sample were separated by SDS-PAGE, using 10% polyacrylamide gels, and were transferred to nitrocellulose membranes. Specific primary antibody (Table 1) and the corresponding IgG-HRP conjugated secondary antibody were used for detection of proteins. The software Image J (http://imagej.nih.gov/ij/ accessed on 4 December 2020) was used for quantification of the intensity of the bands. Values are expressed as the mean ± SEM of normalized values expressed as % vs. cells subjected to hypoxia in the absence of melatonin (non-treated cells).

### 2.11. Quantitative Reverse Transcription-Polymerase Chain Reaction (RT-qPCR) Analysis

RT-qPCR was used to detect the expression of antioxidant enzymes. Cells were incubated under hypoxia with or without melatonin and were then lysed. Lysates were thereafter used for total RNA purification and analysis of protein expression as described previously [15]. Total RNA samples were purified using a commercially available kit (Sigma, Madrid, Spain). The Power SYBR Green RNA-to-CTTM 1-Step kit (Applied Biosystems, Township, USA) was used. Reverse transcription was performed for 30 min at 48 °C, and PCR conditions were 10 min at 95 °C followed by 40 cycles of 15 s at 95 °C plus 1 min at 55 °C using the primers listed in Table 1. The mRNA abundance of each transcript was normalized to the *Gapdh* mRNA abundance obtained in the same sample. The relative mRNA levels were calculated using the ΔΔCt method, and were expressed as the fold change between sample and calibrator. The experiments were carried out employing batches of cells obtained from different preparations. The primers used are listed in Table 2.

RT-qPCR was performed for 30 min at 48 °C. PCR conditions were 10 min at 95 °C followed by 40 cycles of 15 s at 95 °C plus 1 min at 55 °C using the primers listed above (purchased from Thermo Fisher; Madrid, Spain). The abundance of *Gapdh* mRNA in each sample was used for normalization (*n* = 3 independent experiments).

### 2.12. Determination of Cell Viability

Cell viability was studied using crystal violet test, as described previously [7]. Briefly, cells were subjected to different treatments under hypoxia and fixed with 4% paraformaldehyde. After washing, cells were stained with crystal violet (0.1%). Following several washing steps with distilled water, the wells were allowed to air dry and then acetic acid (10%) was added to each well of the plate to dissolve the precipitate. Finally, the absorbance (590 nm) of each sample was measured. A plate reader (CLARIOstar Plus, BMG Labtech., C-Viral, Madrid, Spain) was used. The absorbance was related with the viability of cells subjected to each treatment.

The viability of cells subjected to drugs was compared with that of non-treated cells. Data show the mean change of absorbance expressed in percentage ± S.E.M. (n) with respect to non-treated cells incubated under hypoxia (n is the number of independent experiments).

### 2.13. Statistical Analysis

Data were analyzed for statistics using one-way analysis of variance (ANOVA) followed by Tukey post hoc test, and only *p* values < 0.05 were considered statistically significant. For comparisons and statistics between individual treatments, we employed the Student’s *t* test and only *p* values < 0.05 were considered statistically significant.

## 3. Results

### 3.1. Effect of Melatonin on the Oxidative State of PSC

We were interested in testing whether melatonin evokes ROS production in PSC subjected to hypoxia. In the presence of melatonin, cells exhibited a concentration-dependent change in cytosolic ROS generation. The effect was stronger at the higher concentration of melatonin used (Figure 1A). For comparisons, separate batches of cells were incubated in the presence of hydrogen peroxide (H_2_O_2_, 100 µM) alone, which was used as control of oxidation.

We next evaluated whether treatment with melatonin was accompanied by lipid and/or protein oxidation. Thus, the effects of melatonin on protein carbonyl levels and on TBARS were assayed. In the presence of melatonin, we did not detect increases in the total protein carbonyls content (Figure 1B) nor in the levels of TBARS (Figure 1C). Conversely to what we would have expected, the values detected were lower compared with those detected in non-treated cells, which had been incubated in the absence of melatonin. On its side, treatment of cells with H_2_O_2_ (100 µM) induced statistically significant increases in both total protein carbonyls and TBARS (Figure 1B,C), reflecting an increase in oxidation.

### 3.2. Effect of Melatonin on Nuclear Factor Erythroid 2-Related Factor and Related Antioxidant Enzymes

Next, we analyzed the phosphorylated state of Nrf2 in cells treated with melatonin. In the presence of melatonin, the phosphorylation of Nrf2 was increased in comparison with that detected in its absence (non-stimulated cells; Figure 2A,B).

We next studied the effect of melatonin on the transcriptional activation of Nrf2-related antioxidant enzymes. Treatment of cells with melatonin evoked concentration-dependent increases in the expression of catalytic subunit of glutamate-cysteine ligase (GCLc), catalase (CAT), NAD(P)H quinone oxidoreductase 1 (NQO1), and heme oxygenase-1(HO1), which are antioxidant enzymes regulated by Nrf2 (Figure 2C–F).

### 3.3. Effect of Melatonin on Superoxide Dismutase

First, we tested the effect of melatonin on the protein levels of SOD1 and SOD2, which were analyzed by Western blotting. The expression levels SOD1 and SOD2 were increased in cells incubated in the presence of melatonin, compared with cells that had been incubated in the absence of melatonin (non-treated cells; Figure 3A–C).

In an additional step, we studied the relative mRNA abundance of SOD1 and SOD2 by RT-qPCR. Treatment of cells with melatonin increased the mRNA levels of SOD2 at all four concentrations tested, compared with that noted in cells incubated in its absence (non-treated cells). However, we only detected an increase in the levels of SOD1-mRNA in cells incubated with 1000 µM melatonin with respect to non-treated cells (Figure 3D,E).

### 3.4. Effect of Melatonin on Glutathione and on Total Antioxidant Capacity (TAC)

Having evaluated the effect of melatonin on the above-mentioned antioxidant defenses, it was of interest to analyze the TAC of PSC after treatment with melatonin under hypoxia. The TAC was increased in cells incubated in the presence melatonin, in comparison with that noted in cells incubated in its absence (non-treated cells). The highest values were achieved with 1000 µM and 100 µM melatonin (Figure 4A).

The next part of the experiments was directed to study the effect of melatonin on glutathione. We detected a decrease in the ratio of GSH/GSSG in cells treated with 1000 µM melatonin. On the contrary, GSH/GSSG was increased in cells treated with 10 µM or 1 µM melatonin, in comparison with cells incubated in its absence (non-treated cells; Figure 4B).

### 3.5. Involvement of Protein Kinase C in Melatonin-Induced Changes in Nrf2-Related Antioxidant Enzymes and in SOD

At this stage, we were interested in analyzing the dependency on PKC of melatonin-evoked changes in the antioxidant enzymes that we had observed. Thus, cells were preincubated in the presence of the PKC inhibitor Ro-31-8220 (5 µM), prior to addition of melatonin. Upon addition of melatonin, and in the presence of Ro-31-8220, cells were further incubated during 4 h under hypoxia. The inhibition of PKC led to a decrease in the phosphorylation of Nrf2 and in the detection of SOD1 and SOD2, i.e., PKC inhibition abolished the increases induced by melatonin that we had noted when cells were incubated with melatonin alone (Figure 5).

### 3.6. Effect of Melatonin on Cell Viability

Bearing in mind the increase in ROS production that we had detected in the presence of the high (micro to millimolar) concentrations of melatonin, we were interested in evaluating the effect of the indolamine on the viability of PSC subjected to hypoxia. For this purpose, PSC were incubated under hypoxia during 48 h in the absence (non-treated cells) or in the presence of melatonin. Separate batches of cells were incubated in the presence of Tps (1 µM) alone, which served as control for cell death. A decrease of cell viability was noted with the higher concentrations of melatonin tested (1000 µM and 100 µM). The drop in cell viability (around 14%) was stronger in cells incubated with 1000 µM melatonin. A drop of approximately 5% in cell viability was noted in cells incubated with 100 µM melatonin. Nevertheless, the effect on cell viability was negligible in the presence of 10 µM or 1 µM melatonin. As expected, incubation of PSC in the presence of Tps alone induced a decrease of cell viability, which dropped by 35% (Figure 6).

### 3.7. Effect of Melatonin on Pivotal Members of Inflammation

To investigate whether melatonin modulates major regulators of the intracellular inflammation pathways in PSC, we carried out the next set of experiments. Western blotting was used to detect the phosphorylation of NF-kB and of the regulatory protein nuclear factor of kappa light polypeptide gene enhancer in B-cells inhibitor-alpha (IκBα). The expression of cyclooxygenase-2 (COX-2) also was analyzed by Western blotting. In the presence of melatonin, a decrease in the phosphorylation of NF-kB was observed, compared with non-treated cells (incubated under hypoxia and in the absence of melatonin). The effect was stronger at the concentration of 1000 µM melatonin (Figure 7A,B). Accordingly, a decrease in the phosphorylation of IκBα was detected (Figure 7A,C).

In a separate set of experiments, we assayed the effect of melatonin on COX-2 expression. In the presence of melatonin, a decrease in the detection of COX-2 was noted, compared with non-treated cells (incubated in the absence of melatonin but under hypoxia). Similar to what had been observed with NF-kB and IκBα, the stronger effect was achieved with the concentration of 1000 µM melatonin (Figure 7A,D).

In the next set of experiments, we evaluated the effect of melatonin on the expression of two major pro-inflammatory interleukins, interleukin-6 (IL-6) and tumor necrosis factor-α (TNF-α). For this purpose, PSC were incubated during 24 h under hypoxia and in the presence of melatonin. Separate batches of cells were incubated in the absence of melatonin and under hypoxia (non-treated cells). The mRNA levels of IL-6 and TNF-α were studied by RT-qPCR. The analysis of the samples revealed a decrease in the mRNA levels of IL-6 in cells treated with 1000 µM melatonin, in comparison with non-treated cells (Figure 7E). With respect to TNF-α, a decrease in the mRNA levels was noted in cells incubated with 1000 µM and 100 µM melatonin, in comparison with non-treated cells (Figure 7F).

## 4. Discussion

Hypoxia is a condition that develops in the tumor microenvironment. It is a consequence of the uncontrolled proliferation of cells within the malignant tissue [5]. Hypoxia has also been related with inflammation and damage to the pancreas [32].

PSC participate in the growth and progression of pancreatic cancer [3,4]. Additionally, PSC might participate in inflammation-associated carcinogenesis [33]. Together with tumor cells, PSC will also be subjected to the low availability of O_2_ existing within the tumor. In order to survive, all cell types included in the tumor will have to adapt to these conditions. In general, the cells present in the mass are able to proliferate under hypoxia and, therefore, contribute to the growth of the cancerous tissue [8].

It is now well accepted that PSC participate in a critical manner in the development of the fibrosis that accompanies the diseases affecting the pancreas [34,35,36]. Moreover, fibrosis and inflammation both contribute to the creation of a microenvironment that allows tumor growth [1]. In this line, the fibrotic tissue within the tumor represents a major target in the treatment of cancer [25] and inflammation [1].

Melatonin reduced the viability of different types of cancer cells [37,38,39], including pancreatic cancer cells [30,40]. Melatonin also diminished the viability of PSC under normoxic conditions [22,41]. In addition, melatonin exerted anti-inflammatory actions in the pancreas [42]. Therefore, it has been postulated that treatment with melatonin might be a promising therapy for the diseases that affect the pancreas.

In the present work, we have studied the early events that occur in PSC when subjected to pharmacological concentrations of melatonin under hypoxia. We have shown that, in the presence of melatonin, PSC exhibited an initial production of ROS that might lead to the modulation of their antioxidant responses. This would protect the cells against the putative pro-oxidant conditions that we have recently related with the increased proliferation of PSC under hypoxia [7]. Consequently, these changes could influence their viability. Moreover, melatonin modulated the expression of major proteins that are involved in inflammation. This is of major interest, because it has been suggested that the microenvironment has the ability to influence tumor cells behavior. Moreover, it has been proposed that modulation of the physiology of cells that make up the stroma, rather than their elimination, might be effective in the treatment of inflammation and cancer [43].

Former works have suggested that pharmacological concentrations of melatonin (micromolar to millimolar range) induced the generation of ROS in different cellular types, including the pancreas. This effect has been related with a putative modulation of cell viability, majorly of tumor cells [30,44,45,46,47]. In a former work, we showed that melatonin stimulated ROS production in the cytosol in PSC under normoxia [29]. Additionally, previous results of our laboratory showed that hypoxia induced a prooxidant environment in PSC, without detection of cytosolic ROS generation [7]. Our results have shown that PSC, incubated under hypoxia, exhibited a concentration-dependent increase in ROS production in the presence of melatonin. The effect was stronger at the highest concentration of melatonin used (1000 µM). The ratio of GSH/GSSG, another marker of oxidative state, also was influenced by melatonin treatment in a concentration-dependent manner. A decrease in the availability of GSH was noted in the presence of the higher concentration of melatonin tested. Consistent bibliography reports that melatonin stimulates ROS production and decreases GSH in other cellular models, as we have shown in our work [48,49,50,51,52,53]. On the contrary, GSH/GSSG was increased in cells treated with 10 µM or 1 µM melatonin. Glutathione system plays a pivotal role in the mechanisms that control the oxidative state in the cell, and it is actively consumed in the presence of a pro-oxidant context [54]. In this line, melatonin might behave as a two-faced molecule that, depending on its concentration, could induce pro-oxidant or antioxidant effects. It has also been suggested that different cells may respond to the same concentrations of melatonin differently. In this line, the evidence suggests that the pro-oxidant action of melatonin is not necessarily correlated with cytotoxicity, which is concentration dependent as well as cell type dependent. Thus, how the cells manage melatonin-evoked ROS production will determine cell fate: Survival or death. This also might depend on the cell type and the context [47]. Indeed, previous works have reported pro-oxidant actions of melatonin that could be the basis of its antiproliferative effects in cancer cells [30,55,56]. Conversely, melatonin protected pancreatic acinar cells by increasing their antioxidant defenses [15].

Our results have also shown that, despite the increase observed in ROS generation in response to melatonin, the oxidation of lipids and proteins was kept to a low level. Similar observations were obtained by Orhan et al. [57], who showed that menadione induced the oxidation of fluorescent probes, whereas no increase in the formation of protein oxidation products was observed. Therefore, upon generation or addition of ROS, it might not be necessary to detect oxidation of lipids or proteins in all cell types, despite the fluorescent probes report changes in ROS production. The final consequence will depend on how the cells manage this situation and whether the cells are able to set up antioxidant responses that will cope with the pro-oxidant condition created by a certain stimulus or drug.

In a previous work, we showed that PSC subjected to hypoxia did not exhibit cytosolic ROS production, whereas oxidation of lipids and proteins were detected. Interestingly, under these conditions, PSCs proliferated actively [7]. In the present work, the absence of protein and lipid oxidation could be explained on the basis of a stimulation of the antioxidant responses by melatonin, which would protect cellular structures against oxidation caused by hypoxia.

The transcription factor Nrf2 is an important mediator of the antioxidant response [58]. Its activation is related with the expression of several antioxidant and phase II enzymes, which play pivotal roles in the regulation of the redox status in the cell [59]. As we had observed increases in ROS generation in cells treated with melatonin, which were not accompanied by increases in the oxidation of lipids nor proteins, we examined the effect of melatonin on the antioxidant transcription factor Nrf2 and related antioxidant enzymes, in search of a putative potentiated antioxidant response in the presence of melatonin. In this regard, our results have shown that the phosphorylation of Nrf2 was increased by melatonin treatment. Concomitantly, the levels of the antioxidant enzymes GCLc, catalase, NQO1, and HO1 also were increased. It is well known that Nrf2 is regulated by the red-ox state of the cell [58]. Therefore, the resulting red-ox changes induced by melatonin could be responsible for the activation of the Nrf2-dependent pathway and the increase in the levels of the related antioxidant enzymes that we have noted.

In addition to the above-mentioned antioxidant enzymes, there are other antioxidant systems that control the oxidative state within the cell. SOD enzymes represent an additional set of antioxidant elements that provide defense against free radicals [60]. SOD1 (Cu/Zn SOD) is localized within the cytosol, whereas SOD2 (MnSOD) is localized to the mitochondria [61,62]. In a recent work, we have shown that PSC subjected to hypoxia exhibited an increase in the expression of SOD1 and SOD2 [7]. Thus, it was of interest to study the effect of melatonin on the expression of SOD. In this set of experiments, similar results to those obtained with Nrf2 were observed, i.e., melatonin increased the expression of SOD1 and SOD2. Interestingly, a previous work showed that treatment of PSC with melatonin under normoxic conditions reduced the expression of these key enzymes, which are critical for the detoxification of ROS. This could explain why protein oxidation in PSC treated with melatonin was observed under normoxia, but not under hypoxia [29].

In the presence of melatonin, an increase in TAC was noted. This could be related with the increases in the expression of Nrf2-regulated antioxidant enzymes and of SOD that we have observed. Additionally, we cannot discard that the inherent antioxidant properties of melatonin could be contributing to the increase in TAC that we have noted. Altogether, our results suggest that treatment of PSC subjected to hypoxia with melatonin reinforced their antioxidant defenses.

We have additionally shown that PKC is involved in the actions of melatonin to modulate the expression of antioxidant enzymes under hypoxia. Its inhibition resulted in a decrease in the phosphorylation of Nrf2 and in the detection of SOD1 and SOD2. The involvement of PKC in the antioxidant responses evoked by melatonin in pancreatic acinar cells has been shown previously by our group [15]. Our results provide evidence for the mechanistic action of melatonin to modulate the antioxidant responses in the exocrine pancreas. We cannot exclude that other pathways could be activated, which might lead to an increase in the expression of these antioxidant enzymes [63].

Hypoxia has been signaled as a stimulus for cell proliferation within the pancreatic tumor microenvironment, including PSC [41]. Interestingly, melatonin decreases the viability of pancreatic cancer cells [30] and also of PSC treated under normoxia [22]. We could hypothesize that a certain level of unresolved oxidative stress might be a reason by which PSC are committed to the development of potential fibrosis in the pancreas. We have previously shown that PSC exhibited a pro-oxidant status under hypoxia, which probably led to increases in the expression of Nrf2-dependent antioxidant enzymes and of SOD. However, the activation of these critical antioxidant elements might not be enough to resolve the pro-oxidant conditions created by hypoxia and might result in the activation and proliferation of PSC [7]. In fact, PSC subjected to hypoxia alone exhibited a decrease in TAC in comparison with that noted in cells incubated under normoxia [7]. This might be the reason why, despite the hostile pro-oxidative environment created by hypoxia, PSC exhibited an increase in their proliferation. In the present case, a reinforced increase of the antioxidant defenses by melatonin could tend to counteract the production of ROS in PSC incubated under hypoxia. As a consequence, the higher level of antioxidant enzymes that is achieved in the presence of melatonin could resolve the oxidative stress induced by hypoxia. Under these circumstances, PSC could be subjected to less stress and this could decrease their proliferation (Figure 8). This is a major finding that suggests that melatonin might act as a tuner of PSC viability, avoiding an impairment of their physiology under hypoxia that could lead to uncontrolled proliferation and a concomitant fibrosis within the pancreas.

The last part of our research was directed to study the effect of melatonin on putative inflammatory responses settled under hypoxia. NF-kB signaling is involved in inflammation, fibrogenesis, and cancer development [64,65]. Generally, NF-kB complexes are present in an inactive form in the cytoplasm, where they are repressed by inhibitor proteins like IκBα. Phosphorylation of the latter will allow the activation of NF-kB, which will stimulate the transcription of proinflammatory mediators [66]. On its side, IL-6 and TNF-α are important players of the inflammatory response [67,68]. It has been suggested that melatonin has a beneficial therapeutic value in the treatment of inflammation in different tissues [69,70], including the pancreas [71]. Our present results are in agreement with those previous observations and have shown that melatonin modulated the levels of proinflammatory mediators in PSC subjected to hypoxia. This was reflected as a decrease in the activation of NF-kB, and decreases in the expression of COX-2 and in the transcription on IL-6 and TNF-α. Previous findings reported the involvement of NF-kB and of COX-2 in the inflammatory responses of the pancreas that were observed under simulated hypoxia [72]. The connection between oxidative stress and pancreatitis has been documented [73]. Additionally, inflammation has been related with PSC activation and the development of pancreatic fibrosis [74]. Moreover, it has been suggested that melatonin exhibits anti-inflammatory actions in the pancreas [71]. Our results are thus in agreement with these previous observations and support an anti-inflammatory role for melatonin in PSC under hypoxic conditions.

Considering that melatonin induced a slight drop in PSC proliferation and a decrease in the secretion of pro-inflammatory cytokines, together with the decrease in the levels of TBARS and of carbonyls, and the increase in TAC that we have noted, we could argue that melatonin could lead the cells to a somehow lower activated state, but without reaching complete quiescence. We would like to mention that the process of activation of PSC is very complex and multifaceted. Not all drugs induce changes in all parameters of activation. In the study by Estaras et al. [7], we showed that cell viability increased by 20% in comparison with that of cells incubated in normoxia. Furthermore, the levels of TBARS were increased by 104% and those of carbonyls increased by 67% with respect to the levels detected in cells incubated in normoxia. Additionally, a drop in TAC of 32% was noted. These observations are interesting because our present results are contrary to those observed in PSC subjected to hypoxia only [7] and, hence, might underlie putative pro-quiescency effects of melatonin.

The concentrations of melatonin that we have used in this study could be considered rather pharmacological than physiological, because they are higher than those found normally in blood. Nevertheless, the levels of melatonin found in blood do not necessarily indicate the concentrations of melatonin present in the extracellular space that is immediately close to the cells. This is because it has been suggested that the levels of melatonin in vivo in various body fluids and cells are not necessarily in equilibrium with those detected in blood. In addition, it has been proven that there are tissues that synthesize melatonin, which will act locally, e.g., as an autocrine or paracrine agent. Therefore, values of melatonin that are several orders of magnitude higher than the level found in blood have been found in these tissues [74,75,76,77,78]. Consequently, the levels of melatonin in blood cannot be strictly used to define physiological concentrations, because the local concentrations of melatonin are not defined yet.

## 5. Conclusions

In summary, pharmacological concentrations of melatonin evoked an increase in the production of ROS and a decrease in GSH in PSC subjected to hypoxia. Melatonin treatment led to the activation of antioxidant responses, which involved the expression of Nrf2-regulated antioxidant enzymes and of SOD, and allowed the cells to control the putative pro-oxidant conditions induced by hypoxia. PKC was involved in these responses. Treatment of cells with concentrations of melatonin in the range of millimolar and high micromolar diminished viability of PSC without inducing a massive cell death. In addition, melatonin reduced the activation of the inflammatory pathway and reduced the expression of cytokines (Figure 9). Bearing in mind the pivotal role of PSC in the fibrotic reaction that contributes to survival and development of transformed epithelia within the pancreas, the conditions created by melatonin might restrain PSC proliferation to a certain level to control the fibrotic processes that can evolve under hypoxia. Our results are in line with strategies directed to controlling the growth of fibrotic tissue within tumors, which might help in the treatment of cancer. Additionally, our findings support a probable mechanism by which melatonin modulates fibrosis within the pancreas. Moreover, we have to highlight that the actions of melatonin might be cell- and context-dependent. Additional work will have to be carried out to investigate whether there are other metabolic pathways involved in the antiproliferative and/or antifibrotic actions of melatonin in the pancreas. Finally, in vivo studies are needed in order to corroborate the antifibrotic effects of melatonin that we have observed in vitro.

## Figures and Tables

**Figure 1 antioxidants-10-00577-f001:**
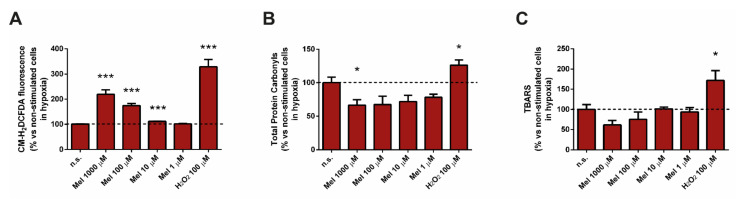
Effect of melatonin on the oxidative state of pancreatic stellate cells (PSC). (**A**) The cytosolic generation of reactive oxygen species (ROS) was evaluated in PSC loaded with the ROS-sensitive dye CM-H_2_DCFDA. (**B**) The effect of melatonin on the levels of protein carbonyls was studied. (**C**) The effect of melatonin on lipid-peroxidation (TBARS) was assayed. H_2_O_2_ (100 µM) was used as a control of oxidation. A horizontal dashed line represents the value observed in cells incubated under hypoxia and in the absence of melatonin (non-treated). Data are representative of three to five independent experiments (n.s., non-treated cells; Mel, melatonin; H_2_O_2_, hydrogen peroxide; *, *p* < 0.05; and ***, *p* < 0.001 vs. non-treated cells in hypoxia).

**Figure 2 antioxidants-10-00577-f002:**
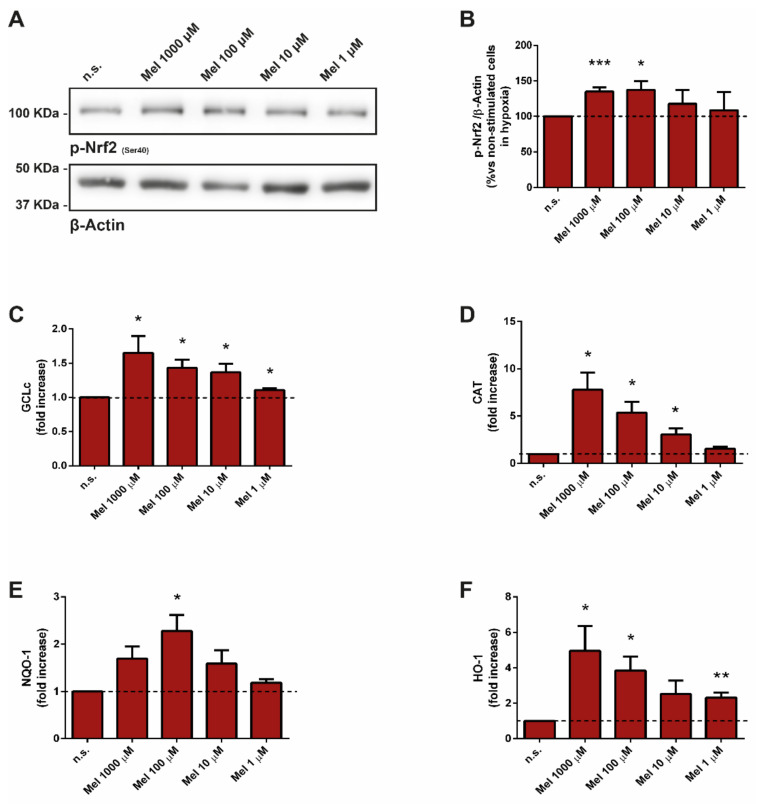
Effect of melatonin on Nrf2-phosphorylation and on Nrf2-regulated enzymes. PSC were incubated with melatonin (1000 µM, 100 µM, 10 µM, or 1 µM) during 4 h under hypoxia. (**A**) Blots showing the effect of melatonin on the phosphorylation of Nrf2. Western blotting analysis was carried out using a specific antibody against the protein. The levels of β-actin were employed as controls to ensure equal loading of proteins. (**B**) The bars show the quantification of protein phosphorylation. A horizontal dashed line represents the value observed in non-treated cells (n.s., incubated under hypoxia but in the absence of melatonin). Results are the mean ± S.E.M. of normalized values, expressed as % with respect to non-treated cells. Four independent experiments were carried out. (**C**–**F**) RT-qPCR analysis was used to study the effect of melatonin on the levels of various Nrf2-dependent antioxidant enzymes: Cysteine ligase-catalytic subunit (**A**; *GClc*), catalase (**B**; *CAT*), NAD(P)H quinone oxidoreductase 1 (**C**; *NQO1*), and heme-oxygenase-1 (**D**; *HO-1*). *Gapdh* mRNA was used for normalization. The bars show the fold increase of mRNA levels ± S.E.M. of each protein relative to non-treated cells (n.s., incubated under hypoxia but in the absence of melatonin). Three different preparations were used (n.s., non-stimulated cells; Mel, melatonin; *, *p* < 0.05; **, *p* < 0.01; and ***, *p* < 0.001 vs. non-treated cells in hypoxia).

**Figure 3 antioxidants-10-00577-f003:**
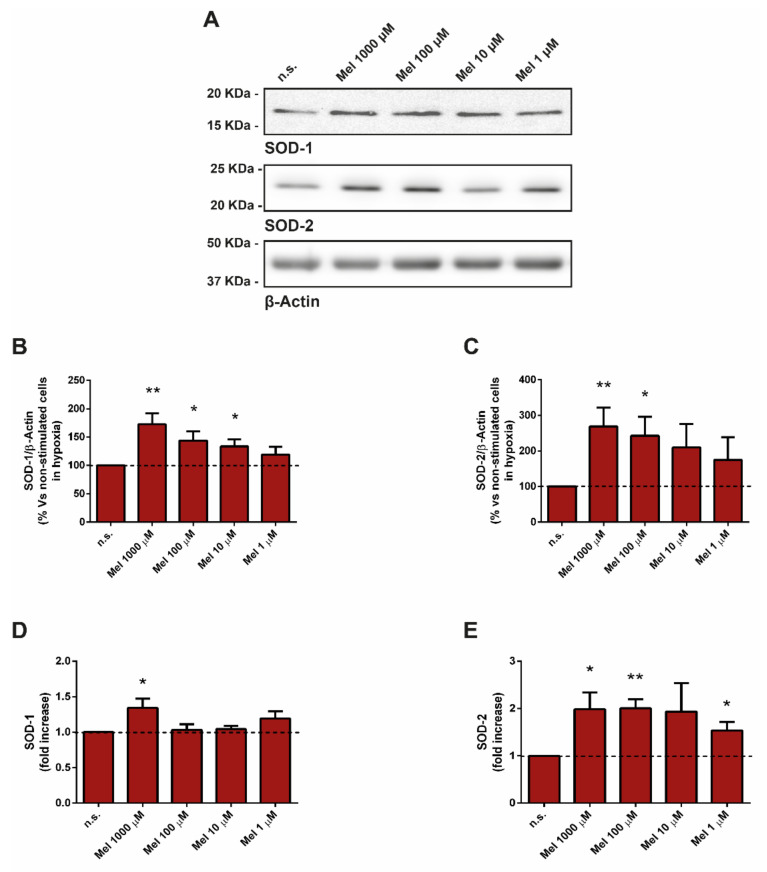
Effect of melatonin on superoxide dismutase (SOD). PSCs were incubated with melatonin (1000 µM, 100 µM, 10 µM, or 1 µM) during 4 h under hypoxia. (**A**) Determination of the expression of SOD1 and SOD2 was carried out by Western blotting, employing specific antibodies. The blots show the effect of melatonin on the expression of these proteins. The levels of β-actin were used as controls to ensure equal loading of proteins. (**B**,**C**) The graphs show the quantification of protein expression. A horizontal dashed line represents the value observed in cells incubated in the absence of melatonin (non-treated). Values are the mean ± S.E.M. of normalized values expressed as % with respect to non-treated cells (incubated under hypoxia and in the absence of melatonin). (**D**,**E**) Detection of mRNA levels of SOD1 and SOD2 was carried out by RT-qPCR analysis. The bars show the fold increase of mRNA levels ± S.E.M. of each protein relative to non-treated cells (incubated under hypoxia and in the absence of melatonin). *Gapdh* mRNA was used for normalization. A horizontal dashed line represents the value observed in non-treated cells. Results are representative of three to four independent experiments (n.s., non-treated; Mel, melatonin; *, *p* < 0.05; and **, *p* < 0.01 vs. non-treated cells in hypoxia).

**Figure 4 antioxidants-10-00577-f004:**
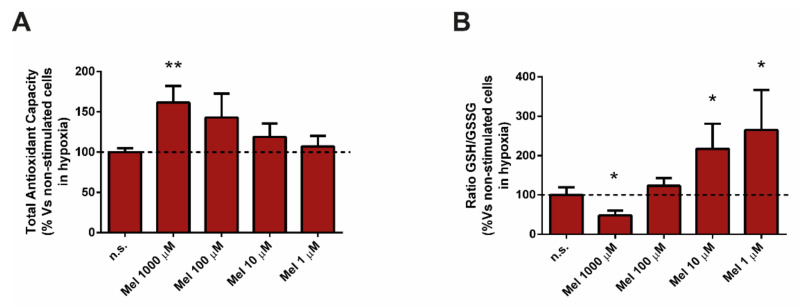
Effect of melatonin on glutathione and on total antioxidant capacity. PSC were incubated with melatonin (1000 µM, 100 µM, 10 µM, or 1 µM) under hypoxia during 4 h. (**A**) Total antioxidant capacity (TAC) of the cells was determined. (**B**) The ratio of GSH/GSSG was studied. Values are the mean ± S.E.M. of normalized values expressed as % with respect to cells incubated under hypoxia, but in the absence of melatonin (non-treated cells). A horizontal dashed line represents the value observed in non-treated cells. Data are representative of five independent experiments (n.s., non-treated cells; Mel, melatonin; H_2_O_2_, hydrogen peroxide; *, *p* < 0.05; and **, *p* < 0.01 vs. non-treated cells in hypoxia).

**Figure 5 antioxidants-10-00577-f005:**
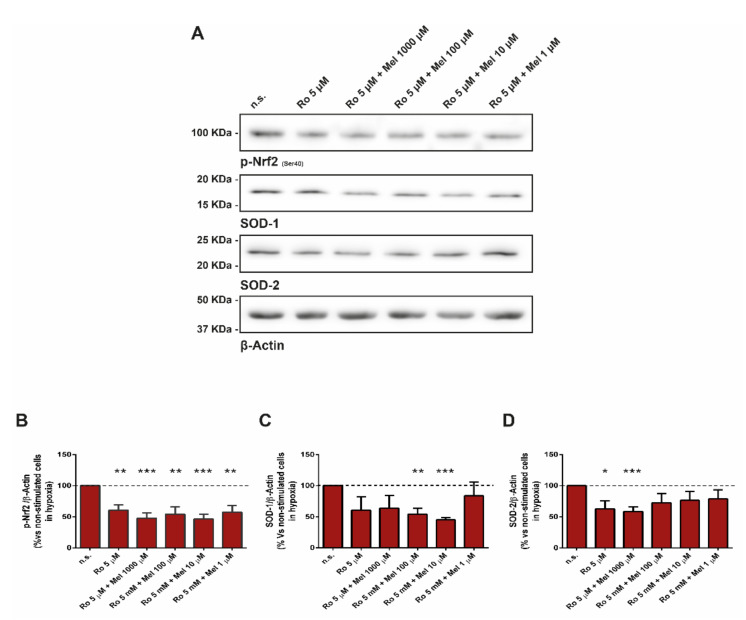
Effect of PKC inhibition on melatonin-evoked changes in Nrf2 and SOD. PSCs were incubated during 5 min in the presence of the PKC inhibitor Ro-31-8220 (5 µM) under hypoxia. Thereafter, melatonin (1000 µM, 100 µM, 10 µM, or 1 µM) was added to the incubation medium and cells were incubated for further 4 h under hypoxia. (**A**) Western blotting analysis was carried out to detect the phosphorylation of Nrf2 and the expression of SOD1 and SOD2 in cells treated with the PKC inhibitor. The levels of β-actin were employed as controls to ensure equal loading of proteins. The blots show the effect of melatonin on protein expression. (**B**–**D**) The graphs show the quantification of Nrf2 phosphorylation and of SOD1 and SOD2 expression. A horizontal dashed line represents the value observed in non-treated cells (incubated in the absence of melatonin). Values are the mean ± S.E.M. of normalized values expressed as % with respect to non-treated cells. Results are representative of four independent experiments (n.s., non-treated cells; Mel, melatonin; Ro, Ro-31-8220; *, *p* < 0.05; **, *p* < 0.01; and ***, *p* < 0.001 vs. non-treated cells in hypoxia).

**Figure 6 antioxidants-10-00577-f006:**
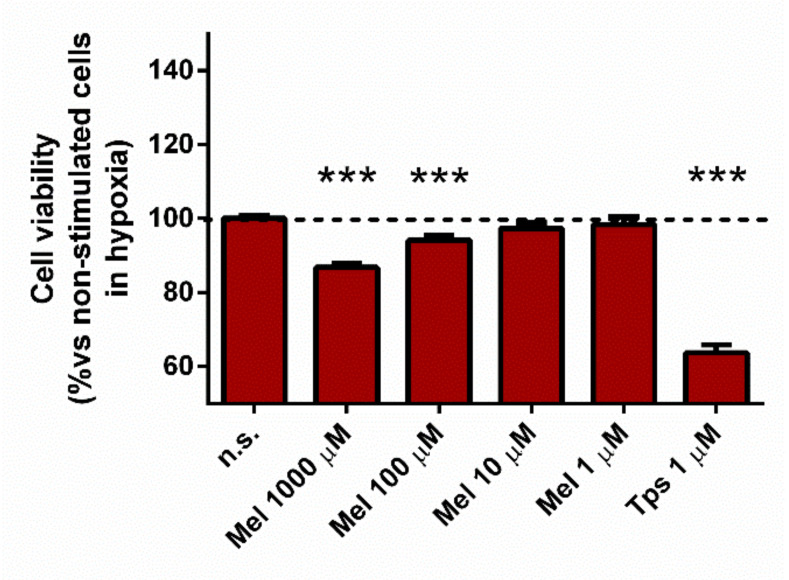
Effect of melatonin on cell viability. Cells were incubated during 48 h under hypoxia and in the presence of different concentrations of melatonin (1000 µM, 100 µM, 10 µM, or 1 µM). Viability was compared with that of cells incubated in the absence of melatonin (non-treated cells). Tps (1 µM) was used as control of cell death. In the graphs, a dashed line represents the viability of non-treated cells. Data are representative of five independent experiments (n.s., non-treated cells; Mel, melatonin; Tps, thapsigargin; ***, *p* < 0.001 vs. non-treated cells).

**Figure 7 antioxidants-10-00577-f007:**
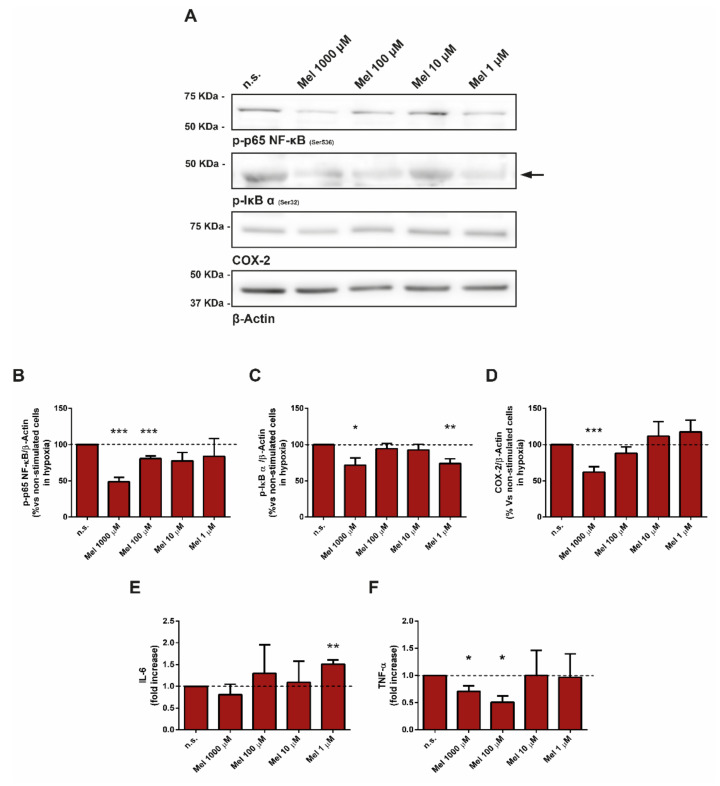
Effect of melatonin on major regulators of inflammation. PSC were incubated with melatonin (1000 µM, 100 µM, 10 µM, or 1 µM) for 4 h under hypoxia. (**A**) Blots showing the effect of melatonin on the phosphorylation state of NF-kB and of IκBα. The expression of COX-2 also was analyzed by Western blotting. The levels of β-actin were employed as controls to ensure equal loading of proteins. The arrow indicates the specific band of IκBα. (**B**,**C**) The bars show the quantification of protein phosphorylation. (**D**) The bars show the quantification of COX-2 detection. Results are the mean ± S.E.M. of normalized values, expressed as % with respect to non-treated cells. Four independent experiments were carried out. (**E**,**F**) RT-qPCR analysis was used to detect the expression of interleuquin-6 (IL-6) and of tumor necrosis factor-α (TNF-α). The bars show the fold increase of mRNA levels ± S.E.M. of each protein relative to non-treated cells (incubated under hypoxia, but in the absence of melatonin). Three different preparations were used. A horizontal dashed line represents the value observed in non-treated cells (incubated under hypoxia but in the absence of melatonin) (n.s., non-treated cells; Mel, melatonin; *, *p* < 0.05; **, *p* < 0.01; and ***, *p* < 0.001 vs. non-treated cells in hypoxia).

**Figure 8 antioxidants-10-00577-f008:**
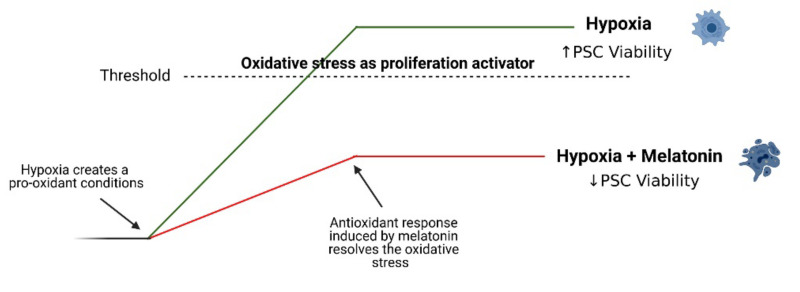
Potential involvement of oxidative stress-related cellular responses in melatonin effect on PSCs physiology. The hostile pro-oxidative environment created by hypoxia could induce a certain level of unresolved oxidative, which might be responsible for the development of potential fibrosis in the pancreas (green trace). On its side, melatonin reinforces the antioxidant defenses that could tend to counteract the production of ROS that we have detected under hypoxia. The consequent higher level of antioxidant enzymes that is achieved could resolve the oxidative stress induced by hypoxia and could decrease the development of fibrosis (red trace). Figure created with BioRender software (BioRender.com accessed on 1 February 2021).

**Figure 9 antioxidants-10-00577-f009:**
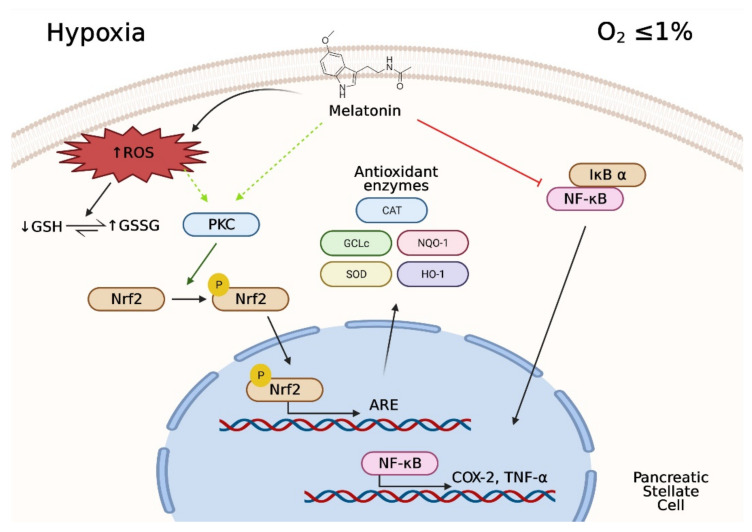
Summary of the major actions of melatonin to modulate PSC responses to hypoxia. The hostile pro-oxidative environment created by hypoxia is modulated by melatonin that, by terms of modulating the antioxidant response element, will lead to diminished oxidative stress. The actions of melatonin also are directed to diminish the inflammatory response settled under hypoxia. These conditions could account for an antifibrotic action of melatonin. Figure created with BioRender software (BioRender.com accessed on 1 February 2021).

**Table 1 antioxidants-10-00577-t001:** Primary antibodies used in the study.

Antibody	Dilution	Supplier
Β-Actin HRP-Conjugated	1:50,000	Thermo Fisher
COX-2	1:2000	Cell Signaling
p-Nrf2 (Ser40)	1:2000	Thermo Fisher
p-p65 NF-κB (Ser536)	1:1000	Cell Signaling
p-IκB α (Ser32)	1:500	Santa Cruz Biotechnology
SOD-1	1:1000	Thermo Fisher
SOD-2	1:2000	Santa Cruz Biotechnology

**Table 2 antioxidants-10-00577-t002:** List of primers used in the study.

Primer	Forward	Reverse
*Cat*	5′-ACTTTGAGGTCACCCACGAT-3′	5′-AACGGCAATAGGGGTCCTCTT-3′
*Gapdh*	5′-GGGTGTGAACCACGAGAAAT-3′	5′-CCTTCCACGATGCCAAAGTT-3′
*Gclc*	5′-GGCACAAGGACGTGCTCAAGT-3′	5′-TGCAGAGTTTCAAGAACATCG-3′
*IL-6*	5′-GTTTGGAAGCATCCATCATTT-3′	5′-TGGAAATGAGAAAAGAGTTGTG-3′
*Ho-1*	5′-AGCACAGGGTGACAGAAGAG-3′	5′-GAGGGACTCTGGTCTTTGTG-3′
*Nqo-1*	5′-GGGGACATGAACGTCATTCTCT-3′	5′-AAGACCTGGAAGCCACAGAAGC-3′
*Sod-1*	5′-GGGGACAATACACAAGGCTGTA-3′	5′-CAGGTCTCCAACATGCCTCT-3′
*Sod-2*	5′-GTGGAGAACCCAAAGGAGAG-3′	5′-GAACCTTGGACTCCCACAGA-3′
*TNF-α*	5′-CCACCAGTTGGTTGTCTTTG-3′	5′-TAGCCCACGTCGTAGCAAAC-3′

## Data Availability

Data are available from the corresponding author upon reasonably request.

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
