# Peer review of "Melatonin Modulates the Antioxidant Defenses and the Expression of Proinflammatory Mediators in Pancreatic Stellate Cells Subjected to Hypoxia"

_antioxidants, 2021, doi:10.3390/antiox10040577_

Round 1

Reviewer 1 Report

The manuscript “Melatonin Modulates the Antioxidant Defenses and the Expression of Proinflammatory Mediators in Pancreatic Stellate Cells Subjected To Hypoxia by Estaras et al. investigated the role of melatonin in the redox status of pancreatic stellate cells under hypoxia.

The manuscript presents well-designed experiments. Comments are listed below:

ABSTRACT

  1. Avoid the separation titles (background, methods…) in the abstract.

INTRODUCTION

  1. The introduction section is too short. Describe the antioxidant mechanism of melatonin and the gap in knowledge that this manuscript aims to cover.

MATERIALS AND METHODS

  1. Revise all chemical formulas (H2O2, CO2, N2, O2, etc.) (Throughout the manuscript).
  2. Revise °C instead of ºC.
  3. Describe the experimental conditions for melatonin treatment in this section. Change 1mM into 1000 µM. Using the same units will ease the comparisons.
  4. More detailed methods are required. In most cases, the authors refer to a reference giving no or little information.

RESULTS

  1. In this section, the authors should avoid referencing. These sentences should go in the discussion section.
  2. In each subsection, the authors state: “cells were incubated during 1 h under hypoxia and in the presence of varying concentrations of melatonin (1 mM, 100 μM, 10 μM or 1 μM)” or similar. The authors should reduce these by giving a complete description of the treatments in the experimental section. In this part, they can refer to them, but concisely.
  3. I consider that the authors should have introduced one more treatment group: non-treated cells under normoxic conditions Just to control the generation of an oxidative environment and see if melatonin restores or prevents the effects of hypoxia. As the results are presented, it is not easy to quantify these effects.
  4. Regarding ROS production (Figure 1A), I consider ROS content is associated with the other parameters. Conversely, ROS increased by melatonin treatment, while the other suffered positive effects. Did the authors consider it could be a methodological error? Melatonin and its metabolic derivatives (AFMK, AMK, among others) present strong fluorescence. Did the authors check the intrinsic fluorescence of cells before adding the ROS prove?
  5. Figure 1A should be reorganized horizontally instead of vertically.
  6. Figures should be reorganized and placed near to the text that described them.

Hence, this manuscript is not ready to be published in the Antioxidants.

Author Response

Manuscript: ID antioxidants-1130816.

Title: Melatonin modulates the antioxidant defenses and the expression of proinflammatory mediators in pancreatic stellate cells subjected to hypoxia.

Authors: Matias Estaras, Manuel R. Gonzalez-Portillo, Remigio Martinez, Alfredo Garcia, Mario Estevez, Miguel Fernandez-Bermejo, Jose M. Mateos, Daniel Vara, Gerardo Blanco-Fernández, Diego Lopez-Guerra, Vicente Roncero, Gines M. Salido and Antonio Gonzalez.

Reply to the reviewer #1 comments

Thank you very much for reviewing our manuscript, for your suggestions and for drawing our attention to the different points raised.

Comment 1: Avoid the separation titles (background, methods…) in the abstract.

Reply: the titles Background, Methods, Results and Conclusion have been removed from the text in the Abstract.

Comment 2: The introduction section is too short. Describe the antioxidant mechanism of melatonin and the gap in knowledge that this manuscript aims to cover.

Reply: Following the reviewer´s suggestion, we have included description regarding the antioxidant mechanisms of melatonin in the introduction section (lines 68 to 77, and 83 to 889). New bibliography has been cited and included in the reference list. In addition, we have also included details for the gap in knowledge that this manuscript aims to cover (lines 88 to 96).

Comment 3: Revise all chemical formulas (H2O2, CO2, N2, O2, etc.) (Throughout the manuscript).

Reply: Thank you very much for this comment. The chemical formulas for H2O2, CM-H2DCFDA, O2, N2 and CO2, have been revised throughout the text and in the legend to figures.

Comment 4: Revise °C instead of ºC.

Reply: °C has been revised throughout the text (line 150) and in the foot to table 2 (lines 279 and 280).

Comment 5: Describe the experimental conditions for melatonin treatment in this section. Change 1 mM into 1000 µM. Using the same units will ease the comparisons.

Reply: Following the reviewer´s suggestion, we have described the experimental conditions used for melatonin in the Material and Methods section. A new subsection has been created (2.4. Experimental conditions for melatonin treatment), lines 147-154. The rest of subsections have been renumbered accordingly. Additionally, 1 mM has been changed to 1000 µM throughout the text, in the figures and in the legend to figures.

Comment 6: More detailed methods are required. In most cases, the authors refer to a reference giving no or little information.

Reply: Following the reviewer´s suggestion, the methods employed are now described with more details.

Comment 7: In this section, the authors should avoid referencing. These sentences should go in the discussion section.

Reply: Following the reviewer´s suggestion, we have deleted referencing included in the results section. We have included this information in the discussion section where appropriate (lines 670-675, 683-684, 699-701, 715-720, 787-790).

Comment 8: In each subsection, the authors state: “cells were incubated during 1 h under hypoxia and in the presence of varying concentrations of melatonin (1 mM, 100 μM, 10 μM or 1 μM)” or similar. The authors should reduce these by giving a complete description of the treatments in the experimental section. In this part, they can refer to them, but concisely.

Reply: Thank you very much for this suggestion. We have deleted this information in the results section. The full description of the treatments is now given in the materials and methods section. A new subsection has been created (2.4. Experimental conditions for melatonin treatment), lines 147-154. The rest of subsections have been renumbered accordingly.

Comment 9: I consider that the authors should have introduced one more treatment group: non-treated cells under normoxic conditions Just to control the generation of an oxidative environment and see if melatonin restores or prevents the effects of hypoxia. As the results are presented, it is not easy to quantify these effects.

Reply: The effect of hypoxia on PSC, and its comparison with normoxia, was studied in a recent work by Estaras et al. (Biol Cell, 2020, DOI 10.1111/boc.202000020). There, we showed that the viability of PSC subjected to hypoxia increased by 20% in comparison with that of cells incubated in normoxia. As sign for pro-oxidative conditions under hypoxia, the levels of TBARS were increased by 104% and those of carbonyls increased by 67% with respect to the levels detected in cells incubated in normoxia. Additionally, a drop in TAC of 32% was noted. We cannot include in the present work results that have been already published by us. However, we have used them for discussion.

In the present work, we showed that melatonin induced a drop in cell viability of approximately 14% and 5% at the concentrations of 1000 µM and 100 µM, respectively. Whereas the effects of 10 µM and 1 µM melatonin on cell viability were negligible. This information was mentioned in the submitted version or the manuscript. The analysis of the results reports that the levels of TBARS dropped in the presence of melatonin by 39% and 24% in cells treated respectively with 1000 µM and 100 µM melatonin under hypoxia. The drop was not as strong as the increase in TBARS noted in cells subjected to hypoxia only which, as mentioned above, increased by 104%. With respect to the levels of carbonyls, treatment of cells with melatonin under hypoxia induced a drop of approximately a 34%, 33%, 28% and 23 % for 1000 µM, 100 µM, 10 µM and 1 µM melatonin, respectively. Again, the decrease in the levels of carbonyls might not counteract the increase in carbonyls detected in cells subjected to hypoxia only, which was increased by 67%. Finally, in cells incubated under hypoxia and in the presence of melatonin, the TAC was increased by approximately a 61%, 43%, 18% and 7% in the presence of 1000 µM, 100 µM, 10 µM and 1 µM melatonin, respectively. The increase in TAC that we have noted with the higher concentrations of melatonin tested (1000 µM and 100 µM) could counteract the decrease in TAC observed in cells subjected to hypoxia only which, as mentioned above, dropped approximately a 32%.

We would like to mention that the process of activation of PSC is very complex and multifaceted. Not all activators induce changes in all parameters of activation. Similar effects occur with the inhibitors. Considering that melatonin induced a slight drop in PSC proliferation and in the secretion of pro-inflammatory cytokines, we could argue that melatonin could lead the cells to a somehow less activated state, but without reaching complete quiescence. Future work should be carried out to shed light on the mechanisms and signalling pathways that are regulated by melatonin to modulate the physiology of PSC and to induce a putative return to a quiescent state.

Reference to this comment has been included in the discussion section (lines 812-821). The text is as follows “Considering that melatonin induced a slight drop in PSC proliferation, together with the decrease in the levels of TBARS and of carbonyls, and the increase in TAC that we have noted, we could argue that melatonin could lead the cells to a somehow lower activated state, but without reaching complete quiescence. In the study by Estaras et al. [7], we showed that cell viability increased by 20% in comparison with that of cells incubated in normoxia. Furthermore, the levels of TBARS were increased by 104% and those of carbon-yls increased by 67% with respect to the levels detected in cells incubated in normoxia. Additionally, a drop in TAC of 32% was noted. These observations are interesting because our present results are contrary to those observed in PSC subjected to hypoxia only [7] and, hence, might underlie putative pro-quiescency effects of melatonin”.

Comment 10: Regarding ROS production (Figure 1A), I consider ROS content is associated with the other parameters. Conversely, ROS increased by melatonin treatment, while the other suffered positive effects. Did the authors consider it could be a methodological error? Melatonin and its metabolic derivatives (AFMK, AMK, among others) present strong fluorescence. Did the authors check the intrinsic fluorescence of cells before adding the ROS prove?

Reply: Thank you very much for calling our attention on this matter. We have been working for a long time with the probes used in this manuscript. Formerly, we have checked the possibility of cellular autofluorescence and of the interference of drugs with dye-derived fluorescence, which were not detected. Therefore, we can assure that there is no interference with the determinations in our experimental conditions that could lead to misunderstanding of our results.

Comment 11: Figure 1A should be reorganized horizontally instead of vertically.

Reply: Following the reviewers´ suggestion we have reorganized the figure 1 horizontally.

Comment 12: Figures should be reorganized and placed near to the text that described them.

Reply: Thank you for calling our attention on this point. The figures are now placed near the text where they are cited.

Reviewer 2 Report

The article presents the results of the detailed study of various effects of melatonin on pancreatic stellate cells (PSC) under hypoxia. The study is an extension of authors’ investigation in the field. The performed in vitro assays are relevant for study of targeted biological effects. The obtained results are presented and discussed in clear way.

The conclusions are, although mostly in accordance with the results, too exaggerate. Prior publishing of the article which can be of quite narrow interest for researchers in the field, the authors are suggested to note in Conclusion (and Abstract) that most of the effects have been observed at very high concentrations and a natural endogenous metabolite melatonin has two-phase hermetic response.

The figure captions of Figures 1 (B and C exchange), 5 (B-D) and 7 (D?) should be corrected. One of Figures 8 /9 is redundant and “could lead to a lower proliferative rate”/“These conditions redirect PSC towards a low proliferation rate profile” should be removed from the relevant caption since it has not been demonstrated enough in the study.

Author Response

Manuscript: ID antioxidants-1130816.

Title: Melatonin modulates the antioxidant defenses and the expression of proinflammatory mediators in pancreatic stellate cells subjected to hypoxia.

Authors: Matias Estaras, Manuel R. Gonzalez-Portillo, Remigio Martinez, Alfredo Garcia, Mario Estevez, Miguel Fernandez-Bermejo, Jose M. Mateos, Daniel Vara, Gerardo Blanco-Fernández, Diego Lopez-Guerra, Vicente Roncero, Gines M. Salido and Antonio Gonzalez.

Reply to the reviewer #2 comments

Thank you very much for reviewing our manuscript, for your suggestions and for drawing our attention to the different points raised.

Comment 1: The authors are suggested to note in Conclusion (and Abstract) that most of the effects have been observed at very high concentrations and a natural endogenous metabolite melatonin has two-phase hermetic response.

Reply: Following the reviewers´ suggestion we have included in the abstract (line 41) and discussion section information regarding this observation (lines 822-833 and line 812). New bibliography has been cited and included in the reference list.

Comment 2: The figure captions of Figures 1 (B and C exchange), 5 (B-D) and 7 (D?) should be corrected. One of Figures 8 /9 is redundant and “could lead to a lower proliferative rate”/“These conditions redirect PSC towards a low proliferation rate profile” should be removed from the relevant caption since it has not been demonstrated enough in the study.

Reply: Thank you very much for this comment. Figure captions in figures 1, 5 and 7 have been corrected.

Additionally, the sentences “could lead to a lower proliferative rate”/“These conditions redirect PSC towards a low proliferation rate profile” have been removed from the text (in discussion section, line 748 and 762) and in the legends to figures 8 and 9.

The legend to figure 8 has been rewritten, and it now states “Figure 8. Potential involvement of oxidative stress-related cellular responses in melatonin effect on PSCs physiology. The hostile pro-oxidative environment created by hypoxia could induce a certain level of unresolved oxidative stress, which might be responsible for the development of potential fibrosis in the pancreas (green trace). On its side, melatonin reinforces the antioxidant defenses that could tend to counteract the production of ROS that we have detected under hypoxia. The consequent higher level of antioxidant enzymes that is achieved could resolve the oxidative stress induced by hypoxia and could lead to a decrease the development of fibrosis (red trace). Figure created with BioRender software (BioRender.com)”.

The legend to figure 9 now states “Figure 9. Summary of the major actions of melatonin to modulate PSC responses to hypoxia. The hostile pro-oxidative environment created by hypoxia is modulated by melatonin that, by terms of modulating the antioxidant response element, will lead to diminished oxidative stress. The actions of melatonin also are directed to diminish the inflammatory response settled under hypoxia. These conditions could additionally account for an antifibrotic action of melatonin. Figure created with BioRender software (BioRen-der.com)”.

Reviewer 3 Report

This is an experimental study of the effects of melatonin on cultured rat pancreatic stellate cells subjected to hypoxia. Melatonin increased, generally dose-dependently, some measures of ROS, as well as gene transcripts and/or protein levels of a number of antioxidant enzymes and inflammatory proteins. A discussion of how these various results could be related mechanistically, and how they may be connected to in vivo pancreatic fibrosis and cancer is presented. There is a certain value to such a cell culture study, as it would incrementally add to the perhaps hundreds of previous studies of melatonin's effects on these components but using different cell types under different conditions. A main area of concern is the lack of sufficient specification of experimental conditions -- too much reliance has been placed on referencing descriptions in previous studies. More detail needs to be given. For example: How pure are the stellate cell cultures (do they contain any other types of cells, and was the purity checked?)? In vivo, stellate cells can be quiescent or activated. Was the state of activation of the cells evaluated during the experiments? How? That should be disclosed. Presumably the shock of sudden hypoxia would cause cell activation, but in particular, it might be of interest to know whether or not some conditions tended to revert the cells to a more quiescent state. What were the concentrations of the gas components in the culture chamber? The cell viability measurements show varying degrees of cell loss under different conditions -- was pH measured and/or controlled, and to what value?

Author Response

Manuscript: ID antioxidants-1130816.

Title: Melatonin modulates the antioxidant defenses and the expression of proinflammatory mediators in pancreatic stellate cells subjected to hypoxia.

Authors: Matias Estaras, Manuel R. Gonzalez-Portillo, Remigio Martinez, Alfredo Garcia, Mario Estevez, Miguel Fernandez-Bermejo, Jose M. Mateos, Daniel Vara, Gerardo Blanco-Fernández, Diego Lopez-Guerra, Vicente Roncero, Gines M. Salido and Antonio Gonzalez.

Reply to the reviewer #3 comments

Thank you very much for reviewing our manuscript, for your suggestions and for drawing our attention to the different points raised.

Comment 1: A main area of concern is the lack of sufficient specification of experimental conditions -- too much reliance has been placed on referencing descriptions in previous studies. More detail needs to be given.

Reply: Thank you very much for this comment. We have given more details about the experimental methods in the corresponding section.

Comment 2: How pure are the stellate cell cultures (do they contain any other types of cells, and was the purity checked?)?

Reply: The procedure we have used is a standardized method in or laboratory. With this procedure, an enriched culture of pancreatic stellate cells with no contamination of other cell types is obtained and has been previously checked (Santofimia-Castaño et al., 2015; Estaras et al., 2019). Information regarding this comment has been included in the manuscript (Material and methods section, Culture of pancreatic stellate cells, lines 139-140). Bibliography is provided. In addition, a supplementary file (suppl. Fig. 1) has been prepared, which shows the expression of specific markers for pancreatic stellate cells (α-smooth muscle actin and collagen-1), mentioned in lines 141-142.

Comment 3: In vivo, stellate cells can be quiescent or activated. Was the state of activation of the cells evaluated during the experiments? How? That should be disclosed. Presumably the shock of sudden hypoxia would cause cell activation, but in particular, it might be of interest to know whether or not some conditions tended to revert the cells to a more quiescent state.

Reply: Thank you very much for this comment. With the procedure that we have used, we obtained cultures of activated PSC. This is mentioned in the material and methods section, including bibliography (lines 139 and 140). To check whether PSC in culture are in an activated state, the expression of α-smooth muscle actin was studied. This has been done in the present and in former works, which were carried out in our laboratory. This was also corroborated in the study by Estaras et al. (Biol Cell, 2020, DOI 10.1111/boc.202000020), cited in the manuscript, where we show that PSC exhibit ability to grow and to migrate under hypoxic conditions, which are hallmarks for cell activation. Additionally, we have checked the expression of collagen type-1. A supplementary figure 1 has been prepared, which shows the detection of α-smooth muscle actin and collagen type-1 by confocal microscopy.

With regard to whether or not some conditions tended to revert the cells to a more quiescent state, we must mention that the process of activation of PSC is very complex and multifaceted. Not all activators induce changes in all parameters of activation. Similar effects occur with the inhibitors. Considering that melatonin induced a slight drop in PSC proliferation and in the secretion of pro-inflammatory cytokines, we could argue that melatonin could lead the cells to a somehow less activated state, but without reaching complete quiescence. Future work should be carried out to shed light on the mechanisms and signalling pathways that are regulated by melatonin to modulate the physiology of PSC and to induce a putative return to a quiescent state.

We have included reference to this observation in the discussion section (lines 812-821). The text is as follows “Considering that melatonin induced a slight drop in PSC proliferation, together with the decrease in the levels of TBARS and of carbonyls, and the increase in TAC that we have noted, we could argue that melatonin could lead the cells to a somehow lower activated state, but without reaching complete quiescence. In the study by Estaras et al. [7], we showed that cell viability increased by 20% in comparison with that of cells incubated in normoxia. Furthermore, the levels of TBARS were increased by 104% and those of carbon-yls increased by 67% with respect to the levels detected in cells incubated in normoxia. Additionally, a drop in TAC of 32% was noted. These observations are interesting because our present results are contrary to those observed in PSC subjected to hypoxia only [7] and, hence, might underlie putative pro-quiescency effects of melatonin”.

Comment 4: What were the concentrations of the gas components in the culture chamber?

Reply: The system we used to induce hypoxia comprises of an electronically controlled atmosphere chamber from Okolab (http://www.oko-lab.com/live-cell-imaging/heating-cooling). The system allows control of gas composition accuracy of ±0.1% with adjustable total output flow rate. The composition of the atmosphere was: 1% O2, 5% CO2 and the rest was adjusted with N2 (94%). Temperature was adjusted to 37 ºC and relative humidity was 90%. The composition of the atmosphere is given in the text (Material and methods section, Induction of hypoxia, line 151). These culture conditions have previously been shown to be effective to induce a state of hypoxia. This was confirmed in a previous work in which we detected the presence of HIF-1 and HIF-2 in PSC cultured under the mentioned conditions (Estaras et at., Bio,Cell, 2020, DOI 10.1111/boc.202000020).

Comment 5: The cell viability measurements show varying degrees of cell loss under different conditions -- was pH measured and/or controlled, and to what value?

Reply: Thank you very much for this comment. The culture medium is made on a basis of medium 199, which contains buffers to maintain physiological pH. In addition, the medium contains phenol red, which serves as an indicator for pH changes. Addition of melatonin or other compounds to the culture medium did not change the pH. During the incubation periods, the pH of the medium did not vary and was maintained stable within the physiological range (7.30-7.35). Information regarding this comment has been included in the manuscript (Material and methods section, Induction of hypoxia, line 152-153).

Reviewer 4 Report

Estaras et al. investigated the effect of Melatonin of hypoxic pancreatic stellate cells. They assessed ROS production, Nrf2-mediated antioxidant defence, cellular antioxidant capacity, involvement of PKC, viability and NFκB signalling. Unfortunately, the results are too preliminary, and shortcomings in the experimental design prevented understanding of their significance.

Major points:

1/ The English of the manuscript is poor, although not uncomprehensible.  A proofing by a native speaker scientist seems necessary. This reviewer stopped listing the mistakes and unclear sentences after line 114.

2/ The positive control H2O2 increased ROS production the same way as melatonin did (Fig 1A). In contrast, melatonin inhibited lipid peroxidation and protein oxidation in a concentration dependent manner, while H2O2 increased both of them (Fig 2B,C). This discrepancy has not been experimentally resolved, therefore, the explanation the authors provided seems unsupported.

3/ The authors should have used H2O2 as positive control in all their experiments, not just in Fig 1, if they wanted to study melatonin’s antioxidant effects. In this respect, it is entirely unclear why they used thapsigargin as a positive control in the cell viability experiments. Thapsigargin inhibits SERCA thereby depletes lumenar Ca2+ stores that affects intracellular ROS production only in a very roundabout way.

4/ In all experiments, the effect of melatonin was compared with untreated hypoxic cells. To understand the results, it would have been important to know the effects of hypoxia on the measured parameters in comparison with untreated normoxic PSC.

5/ Therapeutic concentration of melatonin is about 20 μM. Accordingly, the concentration point of 1 mM seems entirely irrelevant.

Minor points:

Lines 53-55: Clarify the sentence. Do you mean ...adaptation to existence at low oxygen availability...?

Line 72: Instead of “hypothetically” try something like … in order to demonstrate its therapeutic potential in the treatment of pancreatic inflammation and cancer.

Line 93: Is dilution of Β-Actin HRP-Conjugated 1:50.000 or 1:50,000?

Lines 95-98: Quite confusing. Please clarify corresponding. Is it HRP-conjugated species specific secondary antibody?

Line 114: Is it .. incubated with the indicated compounds for 1 h? Stimuli and during  are misused throughout the text.

Lines 323-330: Reference for Fig 1B and C are replaced in the legend.

Author Response

Manuscript: ID antioxidants-1130816.

Title: Melatonin modulates the antioxidant defenses and the expression of proinflammatory mediators in pancreatic stellate cells subjected to hypoxia.

Authors: Matias Estaras, Manuel R. Gonzalez-Portillo, Remigio Martinez, Alfredo Garcia, Mario Estevez, Miguel Fernandez-Bermejo, Jose M. Mateos, Daniel Vara, Gerardo Blanco-Fernández, Diego Lopez-Guerra, Vicente Roncero, Gines M. Salido and Antonio Gonzalez.

Reply to the reviewer #4 comments

Thank you very much for reviewing our manuscript, for your suggestions and for drawing our attention to the different points raised.

Comment 1: The English of the manuscript is poor, although not uncomprehensible.  A proofing by a native speaker scientist seems necessary. This reviewer stopped listing the mistakes and unclear sentences after line 114.

Reply: The English of the manuscript has been checked and corrected.

Comment 2: The positive control H2O2 increased ROS production the same way as melatonin did (Fig 1A). In contrast, melatonin inhibited lipid peroxidation and protein oxidation in a concentration dependent manner, while H2O2 increased both of them (Fig 2B,C). This discrepancy has not been experimentally resolved, therefore, the explanation the authors provided seems unsupported.

Reply: Thank you very much for calling our attention on this point. H2O2 was used as a control to test whether the fluorescent probe CMH2DCFDA was sensitive to ROS. H2O2 is an oxidant and will lead to direct oxidation of the probe. This was detected as an increase in fluorescence. In addition, H2O2 will induce the oxidation of cellular components, as for example lipids and proteins, as we have shown. On its side, melatonin induced an increase in ROS production that was not accompanied by increases in the oxidation of lipids and proteins. This is of relevance because, despite the increase in ROS that we noted, the absence of oxidation of lipids and proteins could be explained by the activation of cellular antioxidant defenses in response to melatonin. Comment to this has been given in the discussion section (lines 695-698): “In the present work, the absence of protein and lipid oxidation could be explained on the basis of the stimulation of the antioxidant responses by melatonin, which would protect cellular structures against oxidation caused by hypoxia”.

Comment 3: The authors should have used H2O2 as positive control in all their experiments, not just in Fig 1, if they wanted to study melatonin’s antioxidant effects. In this respect, it is entirely unclear why they used thapsigargin as a positive control in the cell viability experiments. Thapsigargin inhibits SERCA thereby depletes lumenar Ca2+ stores that affects intracellular ROS production only in a very roundabout way.

Reply: Thank you very much for this comment. In our experimental protocols, we used the positive control that we considered more appropriate to illustrate the effect we were looking for. To detect oxidation of the ROS-sensitive probe, as well as oxidation of lipids and/or proteins, we used H2O2, which is a well-known oxidant. We, and others, have successfully used H2O2 as positive control in studies of oxidative stress. In the studies of cell viability, we used thapsigargin as a control for cell death. Thapsigargin is a recognized and well-known inducer of cell death that has been used extensively in cell viability studies. Both, H2O2 and thapsigargin, are compounds with which we have vast experience. Summarizing, each biological parameter needs its appropriate control. From our experience, we can say that is difficult to use a one-for-all drug as control in the laboratory; i.e., each study of biomarkers will need its control of choice, which does not necessarily need to be the same compound. This is the reason why we used H2O2 as a control for oxidation and thapsigargin as a control for cell death.

Comment 4: In all experiments, the effect of melatonin was compared with untreated hypoxic cells. To understand the results, it would have been important to know the effects of hypoxia on the measured parameters in comparison with untreated normoxic PSC.

Reply: The effect of hypoxia on PSC, and its comparison with normoxia, was studied in a recent work by Estaras et al. (Biol Cell, 2020, DOI 10.1111/boc.202000020). There, we showed that the viability of PSC subjected to hypoxia increased by 20% in comparison with that of cells incubated in normoxia. As sign for pro-oxidative conditions under hypoxia, the levels of TBARS were increased by 104% and those of carbonyls increased by 67% with respect to the levels detected in cells incubated in normoxia. Additionally, a drop in TAC of 32% was noted. We cannot include in the present work results that have been already published by us. However, we have mentioned and discussed these previous results in the revised version of the manuscript (lines 812-821).

Comment 5: Therapeutic concentration of melatonin is about 20 μM. Accordingly, the concentration point of 1 mM seems entirely irrelevant.

Reply: Thank you very much for this observation. The effects have been observed at high concentrations of melatonin, which are considered pharmacological. We have included in the abstract (line 41) and discussion section information regarding this matter (lines 822-833 and line 835). New bibliography has been cited and included in the reference list.

Comment 6: Lines 53-55: Clarify the sentence. Do you mean ...adaptation to existence at low oxygen availability...?

Reply: This sentence (now lines 57-59) has been rewritten to avoid misunderstanding. It now states: “The cells comprising of the mass exhibit adaptation to the low oxygen availability and set up different mechanisms that will help them survive. These changes allow the growth of the tumor”.

Comment 7: Line 72: Instead of “hypothetically” try something like … in order to demonstrate its therapeutic potential in the treatment of pancreatic inflammation and cancer.

Reply: We appreciate this comment. The sentence has been rewritten as suggested (now lines 95-97).

Comment 8: Line 93: Is dilution of Β-Actin HRP-Conjugated 1:50.000 or 1:50,000?

Reply: The dilution employed for this antibody was 1:50000. This information has been corrected in the table.

Comment 9: Lines 95-98: Quite confusing. Please clarify corresponding. Is it HRP-conjugated species specific secondary antibody?

Reply: The reviewer is right; this information was not clear in the text. The column for “Specie”, regarding the primary antibody, has been deleted in table 1. In addition, the footnote has been corrected. In the text it states now “The primary antibodies used in the study are listed in table 1. The corresponding HRP-conjugated species-specific secondary antibody was employed” (lines 114-116). With regard to the table, we have now written in the footnote: “The primary antibodies listed were specific for each protein. The detection of the desired protein was carried out by Western blotting analysis, as described in Methods section. Thermo Fisher (Madrid, Spain); Santa Cruz Biotechnology (Quimigen S.L., Madrid, Spain); Cell Signaling (C-Viral, Madrid, Spain).” (lines 120-123).

Comment 10: Line 114: Is it .. incubated with the indicated compounds for 1 h? Stimuli and during  are misused throughout the text.

Reply: Thank you very much for this comment. We have rewritten the sentence. It now states “Cells were subjected to hypoxia and incubated with the indicated compounds for 1 h.” (now lines 167-168). In addition, to avoid repetition throughout the text, we have described the experimental conditions used for melatonin in a subsection in the Material and Methods section (2.4. Experimental conditions for melatonin treatment), lines 155-163. The rest of subsections have been renumbered accordingly.

Comment 11: Lines 323-330: Reference for Fig 1B and C are replaced in the legend.

Reply: The references for Fig 1B and C have been corrected.

Round 2

Reviewer 1 Report

The authors modified their manuscript following the comments of the four reviewers. They properly addreses all the comments and included more information in their paper to support their hypothesis and results. I consider that the paper can now be published.

Author Response

Manuscript: ID antioxidants-1130816.

Title: Melatonin modulates the antioxidant defenses and the expression of proinflammatory mediators in pancreatic stellate cells subjected to hypoxia.

Authors: Matias Estaras, Manuel R. Gonzalez-Portillo, Remigio Martinez, Alfredo Garcia, Mario Estevez, Miguel Fernandez-Bermejo, Jose M. Mateos, Daniel Vara, Gerardo Blanco-Fernández, Diego Lopez-Guerra, Vicente Roncero, Gines M. Salido and Antonio Gonzalez.

Reply to the reviewer #1 comments

Thank you very much for reviewing our manuscript, for your suggestions and for drawing our attention to the different points raised, which helped us to improve the quality of our work.

Comment: The authors modified their manuscript following the comments of the four reviewers. They properly addressed all the comments and included more information in their paper to support their hypothesis and results. I consider that the paper can now be published.

Reply: We appreciate the reviewer´s response and are thankful for the kind attention given to our manuscript.

Reviewer 4 Report

Although the authors addressed some concerns of this reviewer, they could not resolve the problem of ROS production. Melatonin induced cellular ROS formation in a concentration dependent manner (Fig 1A), and at the same time behaved as an antioxidant (rest of the paper). There must be some fundamental problem with the ROS measurement that should be mended. Alternatively, the discrepancy of being on one hand a pro-oxidant and on the other an antioxidant at the same time should be convincingly addressed by adequate experiments.

Ad Comment 2: This reviewer was not convinced by the authors’ responses. After incubating the cells in the presence of H2O2, the medium has to be replaced to a fresh one containing CMH2DCFDA to assess cellular ROS production. Any different experimental setup would detect the sum of externally added and intracellularly produced ROS, which is inappropriate to draw conclusion from.

Additionally, it is extremely unlikely that CMH2DCFDA detects the ROS, but no lipid peroxidation occurs.  Orhan et al. (Application of lipid peroxidation and protein oxidation biomarkers for oxidative damage in mammalian cells. A comparison with two fluorescent probes. Toxicol In Vitro. 2006;20(6):1005-13.) concluded that lipid peroxidation is more sensitive toward ROS than measuring protein oxidation products or oxidation of fluorescent probes.

Ad Comment 3: Thapsigargin induces cell death by triggering ER stress resulting from depletion of intralumenar Ca2+. It is much different from H2O2 induced oxidative cell death. Accordingly, melatonin should differently interact with the mechanisms of the former and the latter. Lysing the cells in hypotonic solution in the presence of a detergent would have caused cell death too, and would not have been an appropriate positive control in a study of antioxidant properties.

Author Response

Rebuttal to reviewer 4.

General comment: Although the authors addressed some concerns of this reviewer, they could not resolve the problem of ROS production. Melatonin induced cellular ROS formation in a concentration dependent manner (Fig 1A), and at the same time behaved as an antioxidant (rest of the paper). There must be some fundamental problem with the ROS measurement that should be mended. Alternatively, the discrepancy of being on one hand a pro-oxidant and on the other an antioxidant at the same time should be convincingly addressed by adequate experiments.

Reply: First of all, we would like to mention that this reviewer raised 11 comments, of which 9 were satisfactorily answered because no observations were received against. The reviewer, however, considers that there are two points (Comment 2 and Comment 3), on which he/she bases his profound criticisms. He/she considers that we did not address these comments satisfactorily, even though we answered and argued in favor of our observations and, further, we included comments in the revised version of our manuscript taking into account the reviewer´s point.

He/she adds now a new comment. The reviewer states that “Melatonin induced cellular ROS formation in a concentration dependent manner (Fig 1A),…There must be some fundamental problem with the ROS measurement that should be mended”. In response to this observation, we must say that it has been suggested that melatonin induces ROS production at pharmacological concentrations (micromolar to millimolar range), which are the concentrations that we have used in our work. However, it has also been suggested that different cells may respond to the same concentrations of melatonin differently. In his line, the evidence suggests that the pro-oxidant action of melatonin is not necessarily correlated with cytotoxicity, which is concentration dependent as well as cell type dependent.

Consistent bibliography exists regarding this point, which reports that melatonin stimulates ROS production and decreases GSH in other cellular models, as we have shown in our work. Some examples are:

Osseni et al. DOI: 10.1016/s0024-3205(00)00955-3, performed in HepG2 cells.

Wölfler et al. DOI: 10.1016/s0014-5793(01)02680-1, carried out in Jukart cells.

Albertini et al. DOI: 10.1196/annals.1378.050, performed using U937 cells.

Bejarano et al. DOI: 10.1111/j.1742-7843.2010.00619.x, which was caried out in Hematopoietic cell lines.

Zhang et al. DOI: 10.1111/j.1600-079X.2010.00815.x, performed in mesangial cells.

Girish et al. DOI: 10.1016/j.bbrc.2013.07.053, carried out using Human platelets.

Various mechanisms have been proposed for ROS generation by melatonin in different cell types. However, this was not the scope of this manuscript, although it deserves future research.

Information regarding this reply has been included in the abstract (line 41) and in the discussion of the manuscript (page 14, 6th paragraph, line 516, lines 526-527 and lines 538-539). The mentioned references have been included in the revised version of the manuscript (references 48-53).

With regard to “the discrepancy of being on one hand a pro-oxidant and on the other an antioxidant at the same time should be convincingly addressed by adequate experiments” we must say that how the cells manage melatonin-evoked ROS production will determine cell fate, survival or death. Moreover, the result is cell- and context-dependent, as mentioned in: Hong-Mei Zhang and Yiqiang. Zhang. Melatonin: a well-documented antioxidant with conditional pro-oxidant actions. J. Pineal Res. 2014; 57:131–146. Doi:10.1111/jpi.12162. The biological relevance of this behavior remains to be determined, i.e., how the cells manage melatonin-evoked ROS production in order to determine cell fate. This was the aim of our work. We have studied early events that occur in our cellular model when subjected to melatonin. The initial production of ROS in the presence of melatonin might turn on the antioxidant responses that we have observed. The latter would protect the cells against a putative prooxidant condition that we have recently related with the increased proliferation that stellate cells exhibit under hypoxia (Estaras et al., 2020, DOI: 10.1111/boc.202000020). Our manuscript is mentioned and used for discussion in the submitted work.

Information regarding this reply has been included in the discussion of the manuscript (page 14, 5th paragraph, lines 515-520; and line 548 of page 15). New bibliography has been cited and included in the reference list (reference number 47).

Ad Comment 2: This reviewer was not convinced by the authors’ responses. After incubating the cells in the presence of H2O2, the medium has to be replaced to a fresh one containing CMH2DCFDA to assess cellular ROS production. Any different experimental setup would detect the sum of externally added and intracellularly produced ROS, which is inappropriate to draw conclusion from.

Reply: we completely disagree with this observation. The protocol that the reviewer suggests is far away from that given by the manufacturer. The description of the product can be found in the following link: https://www.fishersci.es/shop/products/molecular-probes-cm-h2dcfda-general-oxidative-stress-indicator/11530166?searchHijack=true&searchTerm=CM-H2DCFDA&searchType=RAPID&matchedCatNo=CM-H2DCFDA.

In our work, we have followed the manufacturer´s directions on how to use this fluorescent probe. Directions can be found in the link https://assets.fishersci.com/TFS-Assets/LSG/manuals/mp36103.pdf. In the directions it is stated “to load cells prior to exposing the cells to experimental inducements.” Furthermore, it is suggested by the manufacturer to use H2O2 to a concentration of 100 µM to create positive controls. This is what we did in our experimental procedures. Additionally, we are not studying the effects of H2O2 on ROS in stellate cells but those of melatonin. The reviewer could have misunderstood the protocols.

Information regarding this reply has been included in the manuscript, in materials and methods section (page 4, lines 158-159).

Additionally, we want to highlight that we have used this protocol with success in the past to measure ROS generation in response to CCK (Granados et al., 2004, doi: 10.1016/j.mito.2004.02.003), ethanol (Gonzalez et al., 2006, DOI: 10.1016/j.alcohol.2006.03.002; Gonzalez et al., 2007, doi:10.1016/j.brainres.2007.08.040) or luzindole (Estaras et al., 2019, DOI: 10.1016/j.bbagen.2019.07.016; Estaras et al., 2020, DOI: 10.1002/jat.4018), in different cellular types, including stellate cells. Additionally, the fluorescent probe CM-H2DCFDA has been used for determination of ROS production by melatonin by other researchers. References for this observation are: Keshavarzi et al. DOI: 10.22074/cellj.2018.4860; or Chen et al. DOI: 10.1093/aob/mcx207; to cite some.

Part comment “Any different experimental setup would detect the sum of externally added and intracellularly produced ROS”. It might be possible that the reviewer has misunderstood the experimental procedure that we have used or it could be that we need to describe it clearer in the manuscript. In the point 2.4. of the material and methods section, we state “cells were incubated under hypoxia and in the presence of varying concentrations of melatonin (1000 µM, 100 µM, 10 µM or 1 µM), or H2O2 (100 µM) or thapsigargin (Tps; 1 µM).” We want to state clearly that no combination of melatonin with H2O2 or with thapsigargin was added to the cells; i.e., in our experiments, separate batches of cells were incubated with melatonin, or with H2O2 or with thapsigargin.

Of course, the oxidation of CM-H2DCFDA in the control experiments with H2O2 (carried out in cells previously loaded with the fluorescent dye), can be due to the sum of externally added H2O2 and the consequent intracellularly produced ROS in response to H2O2. Nevertheless, we did not add to the cells a combination of H2O2 plus melatonin. The determination of ROS production that we have shown was carried out using separate batches of cells for each treatment; i.e., melatonin was added to a batch of cells different from that to which H2O2 was added. We insist, melatonin was added to the cells in the absence of H2O2. Therefore, the response that we have observed in the presence of melatonin only can be due to the effects of melatonin, because H2O2 was not added to the same batch of cells.

Information regarding this reply has been included in the manuscript, in the materials and methods section (lines 151) and in the results section (lines 291-292; lines 430-431; and lines 436-437).

CommentAdditionally, it is extremely unlikely that CMH2DCFDA detects the ROS, but no lipid peroxidation occurs.  Orhan et al. (Application of lipid peroxidation and protein oxidation biomarkers for oxidative damage in mammalian cells. A comparison with two fluorescent probes. Toxicol In Vitro. 2006;20(6):1005-13.) concluded that lipid peroxidation is more sensitive toward ROS than measuring protein oxidation products or oxidation of fluorescent probes”.

Reply: In the work cited by the reviewer (Orhan et al., 2006) the authors state “Menadione-induced oxidative stress was also confirmed by oxidation of fluorescent probes. However, no increased formation of protein oxidation products was observed.” The reviewer avoided to mention this important fact, highlighted by the authors of the referred manuscript, with which our findings agree. Therefore, in the work by Orhan et al., it was observed with menadione, a well-known oxidant, a similar situation to ours with melatonin. This is, upon generation or addition of ROS it might not be necessary to detect oxidation of lipids or proteins in all cell types, despite the fluorescent probes report changes in ROS production. The final consequence will depend on how the cells manage this situation and whether the cells are able to set up antioxidant responses that will cope with the prooxidant condition created by a certain stimulus or drug. This is our case, which we have described in our manuscript and discussed accordingly and is the major finding of our work.

Information regarding this reply has been included in the discussion section of the manuscript (lines 555-562) The work by Orhan et al. has been cited and included in the bibliography list (reference number 57).

Ad Comment 3: Thapsigargin induces cell death by triggering ER stress resulting from depletion of intralumenar Ca2+. It is much different from H2O2 induced oxidative cell death. Accordingly, melatonin should differently interact with the mechanisms of the former and the latter. Lysing the cells in hypotonic solution in the presence of a detergent would have caused cell death too, and would not have been an appropriate positive control in a study of antioxidant properties.

Part commentThapsigargin induces cell death by triggering ER stress resulting from depletion of intralumenar Ca2+. It is much different from H2O2 induced oxidative cell death.

Reply: We do not agree with the reviewer´s observation. In relation with this comment, we have previously shown that H2O2 releases Ca2+ from intracellular stores, as thapsigargin does (Gonzalez et al., 2004, DOI: 10.1007/s11010-005-3457-6; Granados et al., 2005, DOI: 10.1042/BC20040513). Moreover, like thapsigargin, H2O2 also induces ER stress (Wu et al. 2018, doi: 10.3892/mmr.2018.9443; Yang et al. 2019, doi: 10.2174/1566524019666190415124838; Weng et al. 2017, doi: 10.3892/mmr.2017.6964; Roscoe and Sevier 2020, doi: 10.3390/cells9102314). Therefore, the effects of thapsigargin and H2O2 are not as completely different as the reviewer states and both compounds share common features. Nevertheless, the study of endoplasmic reticulum stress is out of the scope of our work.

Part comment “Accordingly, melatonin should differently interact with the mechanisms of the former and the latter.”

Reply: the reviewer could have misunderstood the purpose of using H2O2 and thapsigargin as positive controls to corroborate ROS production and the induction of cell death, respectively. As argued above, we used H2O2 as positive control for ROS detection. With respect to choosing a positive control for cell death, we selected thapsigargin, which is a well-known cell death-inducer. We are sure that H2O2 at high concentrations will induce cell death. Moreover, we are not studying the interaction of melatonin with the mechanisms activated by H2O2 or by thapsigargin, but the effects of melatonin treatment. The mentioned interaction is out of the scope of our manuscript.

The mechanisms used by melatonin to modulate stellate cells physiology might be different from those used by H2O2 or thapsigargin to induce their effects, as the reviewer mentions. “The mechanisms used by melatonin” is what we studied, and we have found that melatonin potentiates the cellular antioxidant responses involving protein kinase C and diminishes to some extent cell proliferation, which we have reported is increased under hypoxia. This favors the antifibrotic actions of melatonin, bearing in mind the role of stellate cells in fibrosis. This has not been reported for stellate cells yet and is the major contribution of our manuscript.

Information regarding this reply is included in the manuscript, in the discussion section (lines 623-628) and in the conclusion (lines 693-696 of pages 17-18).
